# Towards foundational LiDAR world models with efficient latent flow matching

**Tianran Liu    Shengwen Zhao    Nicholas Rhinehart**

{tianran.liu, jm.zhao}@mail.utoronto.ca    nick.rhinehart@utoronto.ca

University of Toronto

## Abstract

LiDAR-based world models offer more structured and geometry-aware representations than their image-based counterparts. However, existing LiDAR world models are narrowly trained; each model excels only in the domain for which it was built. This raises a critical question: can we develop LiDAR world models that exhibit strong transferability across multiple domains? To answer this, we conduct the first systematic domain transfer study across three demanding scenarios: (i) outdoor to indoor generalization, (ii) sparse- to dense-beam adaptation, and (iii) non-semantic to semantic transfer. Given different amounts of fine-tuning data, our experiments show that a single pretrained model can achieve up to 11% absolute improvement (83% relative) over training from scratch and outperforms training from scratch in 30/36 of our comparisons. This transferability significantly reduces the reliance on manually annotated data for semantic occupancy forecasting: our method exceeds previous baselines with only 5% of the labeled training data of prior work. We also observed inefficiencies of current generative-model-based LiDAR world models, mainly through their under-compression of LiDAR data and inefficient training objectives. To address these issues, we propose a latent conditional flow matching (CFM)-based framework that achieves state-of-the-art reconstruction accuracy using only half the training data and a compression ratio 6 times higher than that of prior methods. Our model also achieves SOTA performance on semantic occupancy forecasting while being 1.98x-23x more computationally efficient (a 1.1x-3.9x FPS speedup) than previous methods. Our **project page** contains additional visualizations and released code.

## 1 Introduction

World models enable agents to implicitly learn the dynamics of the environment by predicting future sensory observations, typically through generative models operating in a latent space [40, 33]. Starting from classic motion prediction tasks like optical flow [29, 31], recent advances have extended motion prediction to RGB-video-based forecasting, i.e., RGB world models, which have demonstrated impressive performance in applications such as autonomous driving [41, 16, 47], robotic navigation [4], and other embodied tasks [7, 54, 28, 55, 21, 1, 3]. These models can generate video sequences conditioned on historical frames and sometimes natural language input. While an image-based world model may suffice for repetitive tasks (e.g., learning robotic arm movements), its utility is limited in complex scenarios that demand geometrically structured information, such as autonomous driving. In these scenarios, the lack of explicit semantic and geometric representations restricts practical application, as it necessitates additional steps to extract spatial information.

In contrast to RGB images, LiDAR provides rich geometric structure and implicitly offers a sparse representation for semantic cues about the environment. Unlike dense, pixel-based observations, a

39th Conference on Neural Information Processing Systems (NeurIPS 2025).

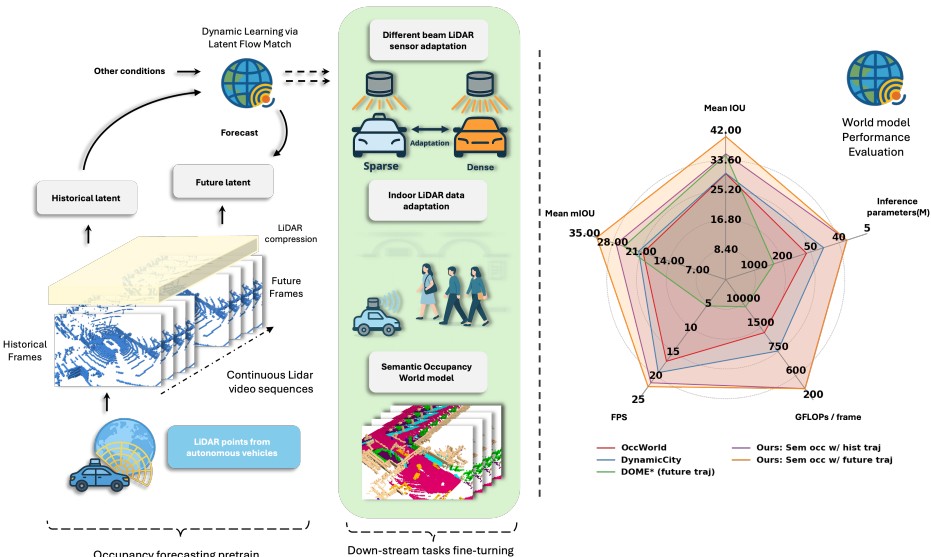

Figure 1: **Left**: The overall pipeline of our method: we used the most readily publicly available LiDAR dataset from autonomous driving scenarios to train the proposed LiDAR world model. This well-trained world model is able to generalize well on the listed downstream tasks after fine-tuning, although scene and signal properties are quite different. **Right:** Comparison of our proposed world model with previous methods on nuScenes 4D semantic occupancy forecasting metrics of mIoU, IoU, and inference efficiency. Our approach achieves the best results in terms of both efficiency and performance.

3D object in a LiDAR point cloud is represented as a cluster of 3D points, encoding essential spatial and semantic information in a compact form. Despite these advantages, foundational world models of LiDAR data remain relatively underexplored: prior LiDAR world models were built for specific domains [56, 20, 57]. In comparison, recent advances in "foundation" models, e.g. pretraining with RGB image forecasting [3] and trajectory forecasting [60] have demonstrated the effectiveness of pretraining in improving performance on downstream tasks. This motivates our central question: Can we develop a foundational LiDAR world model for ground vehicles that yields downstream performance gains on diverse forecasting tasks, particularly those suffering from data limitations, after fine-tuning? This goal is partly motivated by the fact that many of the causal factors that govern the motion of objects are shared across different domains and environments—many aspects of the dynamics of the world should, in principle, be transferable and can be observed in unlabeled LiDAR.

We investigate pretraining and fine-tuning a LiDAR world model across three diverse transfer tasks (Fig 1): varying-beam occupancy forecasting, indoor occupancy forecasting, and semantic occupancy forecasting. First, LiDAR hardware varies in beam count and scan patterns, often degrading model generalization [56, 39]. We consider this cross-sensor setting to assess robustness to hardware variations. Second, robots equipped with LiDAR sensors often operate in vastly different environments [50, 59, 15], from outdoor to indoor settings. While cross-domain adaptation has been extensively studied in perception tasks [24, 34, 30], it remains underexplored in the context of dynamic learning. Given that LiDAR data can be scarce in certain environments (e.g., indoor), it is promising to investigate whether dynamic knowledge learned in data-rich, long-range domains can be transferred to those with limited data and different operational ranges. Lastly, tasks like semantic occupancy forecasting [6, 14, 23, 58]) rely heavily on costly semantic labels [46, 44]. Our goal is to leverage large-scale unlabeled data to learn a universal 3D dynamics prior for ground vehicles, enabling strong semantic forecasting via fine-tuning on minimal labeled data.

Extensive experiments demonstrate that our LiDAR world model significantly improves the convergence speed of downstream tasks. For all mentioned tasks, we observed relative performance gains with varying amounts of fine-tuning data. In particular, for semantic occupancy forecasting, this scheme of learning dynamics prior to semantics patterns allows us to achieve superior performance to OccWorld [58] with only 5% of its required labeled data. Furthermore, our results highlight the importance of representation alignment during fine-tuning. Although the data compression structure we propose later is data-efficient, we found that either using the pretrained data compressor

directly or retraining the VAE from scratch with the fine-tuning data leads to suboptimal performance. This phenomenon is attributable to a feature space mismatch: in both scenarios, the encoder learns a feature space mapping that diverges from the pretrained one, which in turn degrades the performance of the flow model pretrained on the original feature space. We identify two superior strategies: fine-tuning the VAE, and for tasks where the VAE cannot be fine-tuned directly, applying a cosine-similarity-based alignment loss.

We also find that the current architectural paradigms used in LiDAR world models [38, 12] suffer from 2 issues: **redundant model parameters** and **excessive training time**. Regarding the former, the latent representation tends to retain a large number of channels, which significantly increases the parameter count of the dynamics model. Regarding the latter, state-of-the-art models often require thousands of epochs to converge. This inefficiency stems not only from the model scale but also from the inherently slow and compute-intensive nature of denoising diffusion paradigms—particularly those combining DDPM-based training with DDIM-style sampling. These challenges have significantly hindered our exploration of LiDAR world model transferability.

To address these issues, we first propose a Swin Transformer-based VAE architecture for LiDAR data compression. This architecture achieves a compression ratio of 192x—over 6x higher than previous state-of-the-art methods—while matching or exceeding their reconstruction quality. We also propose an efficient flow-matching-based generative model. Compared to previous latent diffusion-based or transformer-based deterministic schemes, our approach requires only 4.38% and 28.91% of the FLOPs that they require, respectively. This efficiency significantly accelerates our analysis of transferability in downstream tasks. In summary, our contributions are:

- The first study on building transferable LiDAR world models: world models of LiDAR videos that exhibit substantial transferability to *downstream forecasting tasks*. We show the efficacy of LiDAR world models to 3 diverse fine-tuning tasks: semantic occupancy forecasting, indoor occupancy forecasting, and high-beams occupancy forecasting, and confirm that it outperforms the baseline of training from scratch on the fine-tuning data, and that the relative performance gain is more pronounced with less fine-tuning data.

- An approach to substantially reduce reliance on human labels for semantic occupancy forecasting: our method exceeds the performance of previous methods with only 5% of the human labels.

- Efficient VAE-baesd architectures for data compression and voxel-based LiDAR world modeling. The former achieves the SOTA reconstruction accuracy at a 6x improvement in compression rate over prior work. Using these encodings, our world model achieves SOTA performance with only 4.47% to 50.23% of the FLOPs and 1.1 to 3.9 times higher FPS.

## 2   Related Work

We categorize previous work by LiDAR-based world modeling for geometric future prediction, semantic 4D occupancy forecasting for semantic-aware scene understanding, and foundational world models (FWMs) that aim to generalize dynamic knowledge across domains and tasks.

**LiDAR-based World Models.** Distinct from general LiDAR generation tasks, LiDAR-based world modeling (also known as LiDAR/Occupancy Forecasting) aims to forecast future sensor observations based on past observation. Occ4D [20] and Occ4cast [27] propose forecasting future LiDAR points or occupancy grids via a differentiable occupancy-to-points module, yet without explicitly modeling latent transition dynamics. UNO [2] further introduces occupancy fields within a NeRF-like [35] framework to enhance forecast fidelity. S2Net [49] adopts a pyramid-LSTM architecture to predict future latents extracted by a variational RNN, while PCPNet [32] leverages range-view semantic maps and a transformer backbone to improve real-time inference performance. Although numerous works [37, 39, 51, 36, 19] have introduced increasingly powerful diffusion-based models, for general data generation, progress in LiDAR forecasting remains comparatively limited compare to the RGB-base ones: Copilot4D [56] achieved state-of-the-art performance in LiDAR forecasting by adopting a MaskGiT-based latent diffusion model with a carefully-designed temporal modeling objective. BEVWorld [57] extended this approach by incorporating multi-modal sensor inputs, enabling future LiDAR prediction even in the absence of current LiDAR frames. However, few works have addressed the *transferability* of dynamics learning across domains—an appealing path to enhance the widespread deployment of world models in real-world autonomous systems.

**Semantic 4D Occupancy Forecasting**: The RGB video-based world models tried to forecast physical consistent video from RGB input. However, these representations lack of geometric and explicit semantic annotation, the usage of the generated forecasts observations in control task still need extra component to recover depth and predict semantics. Semantic 4D occupancy forecasting aims to address this gap by predicting future semantic (LiDAR) occupancy maps based on past observations—either ground-truth annotations or model-generated results. OccWorld [58] employed an auto-regressive transformer to jointly forecast future latent states and corresponding axis offsets. OccSora [45] further advanced the field by being the first to generate 25 seconds semantic videos conditioned on 512x compressed inputs. Later, DynamicCity [6] improved upon this by decomposing the 3D representation into a more compact HexPlane[9] structure, enabling faster inference. Building upon future trajectories/bev layout and extended datasets, DOME [14] and uniScenes [23] continue to push the forecasting accuracy to new state-of-the-art levels. Importantly, these methods rely on labeled semantic ground-truth data for training, making them dependent on expensive human annotations and thus challenging to scale.

**Foundational world model (FWM).** We call a world model "foundational" if it leads to performance gains over learning from scratch on multiple downstream forecasting tasks. In autonomous driving, GAIA-1 [16] and GAIA-2 [41] introduced image-based FWMs that can generate controllable driving scenarios. In the field of indoor robotics, like navigation, NWM [4] proposed an RGB image-based navigation method with a conditional DiT structure. Cosmos [1] takes this ambition further, aiming to generalize across both indoor and outdoor environments. However, due to limitations in the RGB modality, existing FWMs do not provide explicit depth information, which makes it harder to define prediction and planning modules to utilize their output.

## 3 Efficient latent conditional flow matching

In this section, we first introduce our novel point cloud data compressor in Sec.3.1, which achieves state-of-the-art performance under high compression ratio. Based on the compact representation from it, Sec.3.2 presents the flow matching-based generative forecasting model that serves as the testbed for all subsequent fine-tuning approaches. In Sec.3.3, we introduce the VAE fine-tuning for better representation alignment, which will benefit the final forecast performance.

### 3.1 Data compression

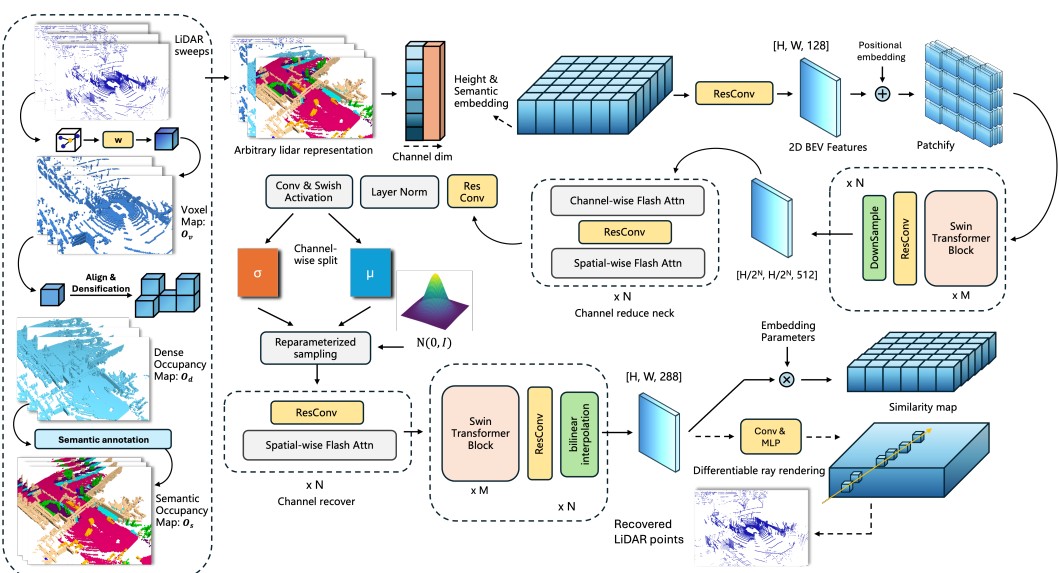

Figure 2: Architecture of our VAE for LiDAR compression. The model enables high compression ratios—exceeding those of previous methods—alongside high-fidelity reconstructions. Here we use $O_v$ to represent the raw voxelized LiDAR points (Occupancy) and $O_d$, $O_s$ stand for densified or semantic labeled occupancy respectively.

While SD3 [12] provides a strong baseline for compressing image-based RGB signals, LiDAR-specific data compression remains underexplored. Existing LiDAR world models typically adopt the SD3 encoder-decoder architecture with minimal modifications, retraining it directly on LiDAR BEV representations. As an alternative, we propose a Swin Transformer-based architecture and demonstrate that it achieves state-of-the-art performance across all forms of LiDAR compression, as detailed in Table 2. As noted in previous works [23], discrete coding-based compression is not only prone to problems such as codebook collapse but is also generally inefficient in compression. Here we have also used continuous coding to achieve a higher compression ratio as shown in Figure 2.

**Data processing and encoding**: In this work, bold notation indicates a multidimensional array (e.g., a vector or a matrix). Given a raw LiDAR sensor scan $\boldsymbol{o}_l \in \mathbb{R}^{L \times 3}$, the voxelized map $\boldsymbol{o}_v$, dense occupancy map $\boldsymbol{o}_d$, and semantic occupancy map $\boldsymbol{o}_s$ can be generated through successive steps of voxelization, densification (i.e., accumulating points from the entire sequence into each frame), and semantic annotation. Our structure is capable of processing any of these representations derived from the raw LiDAR input. Depending on the final reconstruction requirements, voxelization can be either a learning-based approach or a direct binarization process. Taking the compression of $\boldsymbol{o}_d \in \mathbb{R}^{H \times W \times D \times C}$ (a $H, W, D$ 3D grid with C-dim features at each voxel) as an example, a Gaussian encoding distribution $q(\boldsymbol{z}_d | \boldsymbol{o}_d) = \mathcal{N}(\boldsymbol{z}_d; \boldsymbol{\mu}_q, \boldsymbol{\sigma}_q)$ is constructed from three main stages: embedding, feature learning and downsampling, and channel reduction. Specifically, after applying height and class embeddings, a 2D BEV feature map is obtained, which is subsequently processed by a standard 2D Swin Transformer encoder. Unlike the original Swin Transformer design, which utilizes patch merging for downsampling, we replace this operation with conventional convolutional layers—a modification we empirically find to outperform the original approach. Channel reduction is performed by a lightweight network neck, compressing the feature representation into a 16-dimensional latent space. Samples are drawn using the reparameterization trick $\boldsymbol{z} = \boldsymbol{\mu}_q + \boldsymbol{\sigma}_q \odot \boldsymbol{\epsilon}, \boldsymbol{\epsilon} \sim \mathcal{N}(0, \boldsymbol{I})$, and $\odot$ denotes the Hadamard product.

**Decoding and $o_l$ representation recovery.** The decoder $q(\hat{\boldsymbol{o}}_d | \boldsymbol{z}_d)$ is designed to mirror the encoder by starting with the same symmetric 2D block structure. Different from previous methods that used 3D blocks in the decoder to increase temporal feature consistency, we found doing so to have a negative effect in our structure, for both reconstruction and final forecasting result, as shown in the ablation study. Depending on the source representation, the reconstructed $\hat{\boldsymbol{o}}_d$ can be used to render the points with a differentiable ray rendering module or get the occupancy map by calculating a similarity score with the class embedding.

### 3.2 Forecasting with Conditional flow matching

With the proposed VAE, we are able to mitigate the parameter redundancy: most of the parameters of the existing methods come from the high dimensionality of the latents. To further make model training more efficient, we present a new structure based on flow matching [26], which we show leads to SOTA performance in both forecasting accuracy and computational efficiency, as shown in Table 1. For a semantic occupancy forecast task, given $\boldsymbol{o}_s^{t_0:t_2}$, our VAE encodes continuous frames to $\boldsymbol{z}_s^{t_0:t_2} \in \mathbb{R}^{(t_2-t_0) \times H \times W \times 16}$. As $t_1$ is the middle index of these frames, our objective is to obtain the future latent $\boldsymbol{z}_s^{t_1:t_2}$ from the historical latent $\boldsymbol{z}_s^{t_0:t_1}$ using a flow-matching model $G_\theta$.

As illustrated in Figure 3, the training objective is to regress a velocity field $u_t^\theta$. This regression is conditioned on several inputs: a time step $t \in [0, 1]$ representing the progress along a probability path from an initial distribution (standard Gaussian) to the target distribution; the interpolated future latent state $x_t$; the historical trajectory $J^{t_0:t_1} \in \mathbb{R}^{(t_1-t_0) \times 3}$; and historical observations $\boldsymbol{o}_s^{t_0:t_1}$.

Specifically, the noising added follows a linear interpolation method as shown in equation 1, where $\sigma$ used to balance the scale of noise and latents and $\epsilon \sim \mathcal{N}(0, 1)$. The $\boldsymbol{z}_s^{t_0:t_1}$ will be used as a part of condition in training: we concatenate the historical latents and noised future latents along time dimension to get the input of $G_\theta$, denoted by $\mathbf{z}_{\text{joint}} \in \mathbb{R}^{(t_2-t_0) \times H \times W \times 32}$.

$$\mathbf{x}_t = (1 - t)\boldsymbol{\epsilon} + t\sigma \boldsymbol{z}_s^{t_1:t_2} \tag{1}$$

Based on the spatial-temporal DiT structure proposed in previous methods [33, 23, 14], we observe that the convergence speed remains suboptimal. Specifically, given $\mathbf{z}_{\text{joint}}$, the initial 3D convolutional layer increases the channel dimension to $C'$, resulting in $\mathbf{z}'_{\text{joint}} \in \mathbb{R}^{(t_2-t_0) \times H \times W \times C'}$. The width and height dimensions are then flattened before being fed into the spatial DiT, where multi-head attention

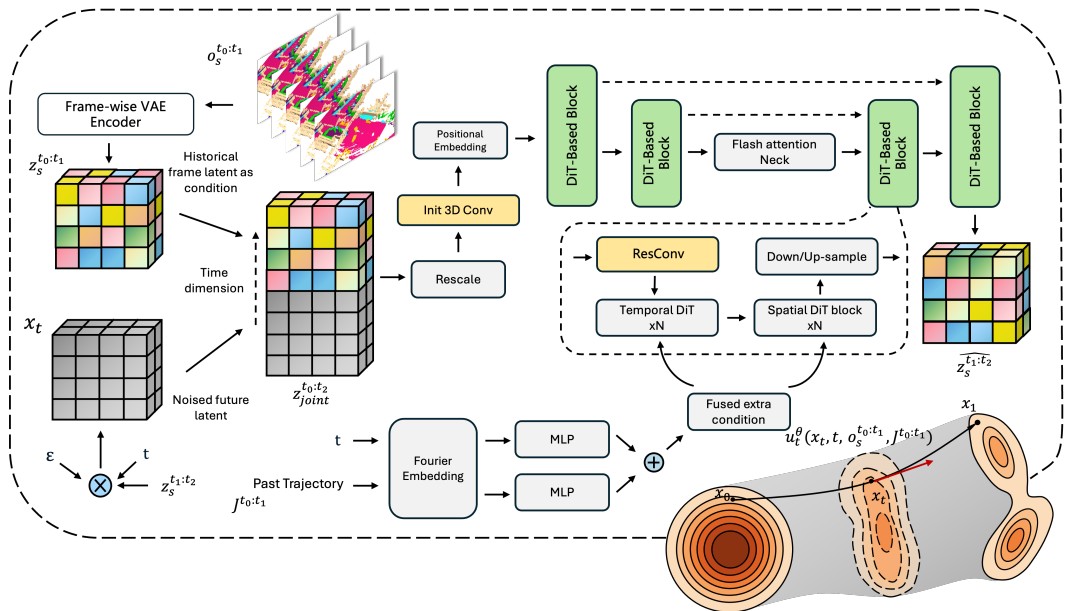

Figure 3: Architecture at the training stage of our proposed conditional velocity field predictor for time $t$. Historical frame latents are extracted via a frame-wise VAE encoder, and noised future (target) latents are formed by injecting noise at timestep $t$. Latents are concatenated along the time dimension and passed through DiT blocks.

(MHA) is applied and normalized using AdaLN. However, applying the same strategy in the temporal DiT would only fuse pixels along the temporal axis, limiting the temporal receptive field to just a single latent pixel(in a block). To compensate, we observed that most of the previous methods like uniScenes [23] and DOME [14] use 14-18 stacked blocks to ensure the temporal consistency, which is also one of the reasons of redundant parameters. While this design remains feasible in Latte [33], which operates on 3D video patches, our per-frame latent representation, making the learning of temporal dependencies more challenging. We found that this issue can be effectively mitigated by simply inserting a 3D convolutional layer with a larger receptive field after the spatial DiT. Furthermore, organizing the network in a UNet-style architecture, as opposed to a single-stride DiT backbone, is shown through later experiments to further enhance the forecast performance and reduce computational load. The training objective was designed following the Rectified Flow [26]:

$$\mathcal{L}(\theta) = \mathbb{E}_{t, \boldsymbol{x}_0, \boldsymbol{z}_s, J} \left\| \mu_t^\theta(\boldsymbol{z}) - (\boldsymbol{z}_s - \boldsymbol{x}_0) \right\|^2, \quad t \sim \texttt{sigmoid}(\mathcal{N}(0, 1)), \; \boldsymbol{x}_0 \sim \mathcal{N}(\mathbf{0}, \boldsymbol{I}), \; \boldsymbol{z}_s \sim q_{\boldsymbol{z}_s}.$$

### 3.3 Improved fine-tuning with representation alignment

The pretrain of world model can now be start with framework proposed in 3.2 & 3.1 and large amounts of unlabeled data $\boldsymbol{o_d}$ in a self-supervised way to learn the unified prior knowledge for environment dynamics. For non-semantic to semantic transfer specifically, we now already have encoder/decoder pair denoted by $q_d(\boldsymbol{z}_d|\boldsymbol{o}_d)$ / $q_d(\hat{\boldsymbol{o}}_d|\boldsymbol{z}_d)$ and forecasting model $G_\theta^d$. Although we can train the other data compressor $q_s(\boldsymbol{z}_s|\boldsymbol{o}_s)$ / $q_s(\hat{\boldsymbol{o}}_s|\boldsymbol{z}_s)$ and use the corresponding latent $\boldsymbol{z}_s$ to fine-tuning pretrained $G_\theta^d$, we have discovered that the latent spaces of $\boldsymbol{z}_s$ and $\boldsymbol{z}_d$ are not aligned, which is affects the performance of fine-tuning.

For subtask 1 and 2 in Fig 1, this suboptimality can be tackled by fine-tuning the data compressor first, while in case of subtask 3 (semantic occupancy forecasting), due to the difference in network embedding layer dimensions, it is non-trivial to fine-tune the semantic data based on a VAE pretrained on non-semantic data. Instead, we use the latents of the corresponding dense occupancy $\boldsymbol{o}_d$ to guide the formulation of the subspace of the semantic VAE.

Specifically, we add a cosine similarity term in the loss when fine-tuning the VAE, as shown in the last term of Equation 2. $\boldsymbol{z}_s$ and $d_s$ stand for the latent from a pair $\boldsymbol{o}_s$ and $\boldsymbol{o}_d$ via training from scratch and pretrained VAE, respectively. $\mathcal{L}_{\text{lovasz}}$ [5] is another reconstruction term used to optimize IoU.

$$\mathcal{L}_{\text{Sem-VAE}} = \mathcal{L}_{\text{CE}}(\hat{\boldsymbol{o}}_s, \boldsymbol{o}_s) + \beta D_{\text{KL}} \left( q_\phi(\mathbf{z}_\mathbf{s} \mid \mathbf{o}_\mathbf{s}) \,\|\, p(\mathbf{z}_\mathbf{s}) \right) + \lambda \mathcal{L}_{\text{lovasz}}(\hat{\boldsymbol{o}}_s, \boldsymbol{o}_s) + \kappa \mathcal{L}_{\cos}(\boldsymbol{z}_s, \boldsymbol{d}_s) \quad (2)$$

Therefore, our full pipeline is 3-fold: we first train the data compressor and flow matching model via the easily accessible unlabeled data. Then we use the pretrained data compressor to guide the structure of latent space for subtask's data. Finally, with the aligned representation, we fine-turn the pretrained weights to obtain the final results.

# 4 Experiments

We designed our experiments to investigate several questions: **Q1**: Can we improve compression and performance over existing designs? **Q2**: Can we develop a LiDAR world model that exhibits substantial superiority to models trained from scratch on three downstream forecasting tasks (high beam occupancy forecasting, indoor occupancy forecasting, and semantic occupancy forecasting)? **Q3**: How well does each fine-tuning variant perform and why?

## 4.1 Experimental design

We use nuScenes[8] (2Hz) as the pretraining data for the flow matching model. We trained 2 different types of foundational models on it, one was trained based on the original LiDAR sweep ($o_v$), totaling 27,000 frames, denoted by $G_\theta^v$. The other type of model was trained using densified LiDAR frames $o_d$), totaling 19,000 frames, denoted by $G_\theta^d$. The latter training set is a subset of the former part and the number of $o_d$ equal to $o_s$.

For the beam adaptation subtask, we down-sampled 11 sequences from the KITTI360 raw dataset [25] from 10Hz to 2Hz to match the foundational model setting. Also, we collected an indoor navigation dataset using a Clearpath Jackal equipped with an OSO-128 LiDAR sensor (training set with 23,504 frames and validation set with 9,720 frames). Finally, for the semantic occupancy forecasting, we present twofold experiments: in section 4.2, we follow the official splitting [44] and train the model from scratch. Then in Section 4.3 and Appendix H.1, we only use first half of the training data to pretrain the $G_\theta^d$ as the foundation model, which then fine-tuning on the other half $o_s$ ($o_s$ and $o_d$ are 1v1 correspondence) to avoid the dynamic knowledge of fine-tuning data to be already seen during pretraining stage. From the next section, we use $o_s'$ and $o_d'$ to denote the partial training data. For more details on the model setup, please refer to our appendix.

## 4.2 Encoder structure exploration and semantic occupancy forecasting evaluation

For model evaluation, Table 2 compare reconstruction results of our Swin-Transformer VAE with previous methods compress data from 8× to 512×. At 32×, we achieve 99.2% mIoU and 97.9% IoU—far above UniScenes's 92.1%/87.0% at the same rate. Even at 192×, we maintain 93.9%/85.8%, surpassing OccWorld and DOME by over 11% mIoU, and at an extreme 768× our model still delivers a 9.7% relative gain despite 1.5× higher compression.

Table 1: Comparison of future occupancy forecasting performance on full nuS. validation set. †: Methods use future trajectory information. ⋆: DynamicCity generates 16 frames jointly with 1000 steps DDPM-style sampling, we report 50 steps DDIM FPS for faster eval but still keep the original accurary. All FPS tested based on 1x RTX4090 without kernel fusion or other CUDA acceleration methods. −: Unreported or unable to be computed due to code unavailability.

| Method | mIoU↑ | | | | IoU↑ | | | | Mean NLL (bits/dim)↓ | Params↓ | GFLOPs per Frame↓ | FPS↑ |
|---|---|---|---|---|---|---|---|---|---|---|---|---|
| | 1s | 2s | 3s | Avg | 1s | 2s | 3s | Avg | | | | |
| OccWorld [58] | 25.75 | 15.14 | 10.51 | 17.13 | 34.63 | 25.07 | 20.19 | 26.63 | – | 72.39 | 1347.09 | 16.97 |
| RenderWorld [52] | 28.69 | 18.89 | 14.83 | 20.80 | 37.74 | 28.41 | 24.08 | 30.08 | – | 416 | – | – |
| OccLLama [48] | 25.05 | 19.49 | 15.26 | 19.93 | 34.56 | 25.83 | 24.41 | 29.17 | – | – | – | – |
| DynamicCity⋆ [6] | 26.18 | 16.94 | – | – | 34.12 | 25.82 | – | – | – | 45.43 | 774.44 | 19.30 |
| **Ours** | **33.17** | **21.09** | **15.64** | **23.33** | **40.53** | **30.37** | **24.44** | **31.78** | 6.29 | **30.37** | **389.46** | **22.22** |
| DOME† [14] | 29.39 | 20.98 | 16.17 | 22.18 | 38.84 | 31.25 | 26.30 | 32.13 | 6.04 | 444.07 | 8891.98 | 5.48 |
| **Ours†** | **36.42** | **27.39** | **21.66** | **28.49** | **43.68** | **36.89** | **31.98** | **37.52** | **4.55** | **30.37** | **389.46** | 21.43 |

Based on the strong VAE performance, our forecasting model advances the state-of-the-art of semantic occupancy forecasting while maintaining real-time throughput. As shown in Table 1, our model achieves a one-second mIoU of 33.17%, surpassing the previous SOTA model's 28.69% and OccLLama's 25.05% by 4.48% and 8.12%

respectively and maintaining 21.09% and 15.64% for two and three second prediction which outpaces prior works by at least 2.5% relative performance improvement.

For future-trajectory-conditioned forecasting, our mIoU surpasses DOME by more than 5.5% across all frames. Our semantic occupancy forecasting model runs at 22.22 FPS, requiring only 389.46 GFlops per frame and 30.37 million parameters. Compared to DynamicCity which generates 16 frames altogether, our model uses half of GFlops per frame, 66% of parameters, and 1.1 times higher FPS. In the future-trajectory-conditioned regime, our model sustains 21.4 FPS without increasing in computational cost and parameter count, whereas DOME utilizes 23 times more flops per frame (our model is 4.38% of this amount), 15 times more parameters, 4 times

Table 2: Occupancy reconstruction performance at various compression ratios.

| Method | Cont.? | Comp. Ratio↑ | mIoU↑ | IoU↑ |
|---|---|---|---|---|
| OccLlama [48] | ✗ | 8 | 75.2 | 63.8 |
| OccWorld [58] | ✗ | 16 | 65.7 | 62.2 |
| OccSora [45] | ✗ | 512 | 27.4 | 37.0 |
| DOME [14] | ✓ | 64 | 83.1 | 77.3 |
| UniScenes [23] | ✓ | 32 | 92.1 | 87.0 |
| UniScenes [23] | ✓ | 512 | 72.9 | 64.1 |
| Ours | ✓ | 32 | **99.2** | **97.9** |
| Ours | ✓ | 192 | 92.8 | 85.8 |
| Ours | ✓ | 384 | 88.3 | 76.9 |
| Ours | ✓ | **768** | 80.0 | 69.3 |

slower FPS, and at least 5.5% less absolute performance. Due to the stochastic nature of our proposed CFM model and the limitation of mIoU and IoU in evaluating model's ability to generate diverse but plausible future predictions, we also report the negative log likelihood (NLL) in bits-per-dimension. Our model also present better performance compare with previous SOTA stochastic models.

In order to further measure sample quality, we computed the 3D Fréchet Inception Distance (FID) and Kernel Inception Distance (KID) in Table 4. Our model establishes a new state of the art on both FID and KID. On average, for our model without future trajectories, it reduces FID to just 28.3 % / 48.5 % and KID to 22.7 % / 44.4 % of the scores reported by the deterministic (OccWorld) and stochastic (DOME) models, respectively. See Appendix A.2.2 for more details of these metrics.

Then, benefiting from the low information loss of the data compressor we proposed, it is in fact highly suitable to be used as feature extractor for 4D inception-based metrics (temporal consistency measurement). After we revise the structure with more temporal (4D) module, we retrain the proposed data compressor to eval the FVD score of generated results. In Table 3, to give context for temporal consistency, we include an approach that randomly shuffles the ground truth future ordering and compare the FVD with the correctly ordered GT. This tells us the FVD when the predictions are individually correct but temporally inconsistent. Our model scores of 7.68 and 7.80 are also significantly lower than previous methods.

Table 3: FVD Score Comparison, 3s (6 frames) videos used as input

| Method | FVD ↓ ($\times 10^{-3}$) |
|---|---|
| OceWorld | 18.68 |
| DOME | 9.79 |
| Ours (hist. traj.) | 7.80 |
| Ours (fut. traj.) | **7.68** |
| Reorder GT | 12.07 |

Table 4: FID and KID($\times 10^{-2}$) comparison for our method and other generative-based and deterministic-based models.

| Method | 1s | | 2s | | 3s | | AVG | |
|---|---|---|---|---|---|---|---|---|
| | FID | KID | FID | KID | FID | KID | FID | KID |
| OccWorld | 9.54 | 11.7 | 8.56 | 10.46 | 7.78 | 9.80 | 8.62 | 10.60 |
| DOME | 4.37 | 4.39 | 5.09 | 5.44 | 5.63 | 6.53 | 5.03 | 5.42 |
| Ours (Hist. traj) | 1.67 | 1.49 | 2.81 | 2.85 | 4.02 | 4.74 | 2.83 | 3.02 |
| Ours (Fut. traj) | 1.61 | 1.47 | 2.90 | 3.05 | 3.91 | 4.55 | 2.80 | 3.02 |

## 4.3 LiDAR world model transferability

First, In Table 5, we compare the pretrained model with previous 4D forecasting (occupancy forecasting) methods [53, 20]. Ours exhibits better performance in non-semantic occupancy forecasting, particularly at the 3-second horizon. Then, using the architecture in Section 4.2, we evaluate the performance gains from fine-tuning our world model with varying strategies and data scales across individual tasks and present the results in Figure 4.

For the high-beam LiDAR adaptation task, all configurations based on the pretrained world model outperform from-scratch VAE and CFM training (blue line) across all frames and data fractions. Pretraining CFM alone yields 17.11% and 6.68% gains for 1s and 3s predictions at 10% data. With both components pretrained and fine-tuned, the gains increase to 14.48% and 26%, respectively. This demonstrates that our Swin Transformer-based VAE effectively captures transferable geometric priors for downstream tasks with varying LiDAR characteristics and the effectiveness of the feature alignment. For the indoor LiDAR adaptation task, all pretrained model variants outperform from-

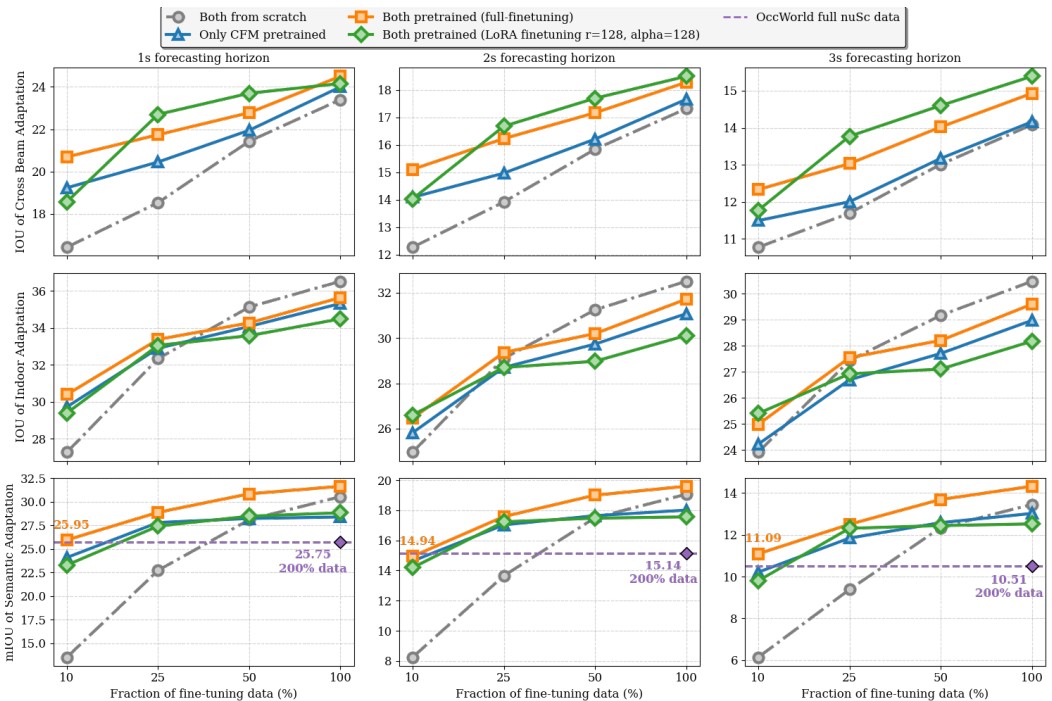

Figure 4: IoU/mIoU comparison across 3s forecasting horizon of the presence of different fraction of fine-tuning data used from total data available between the various training procedures. In row order, each row refers to the results of (i) *different beam adaptation* (ii) *outdoor-indoor adaptation*, and (iii) *semantic occupancy forecasting*, respectively.

scratch training when fine-tuning data is limited (less than 25%), with full-parameter tuning of both VAE and CFM achieving the best results. At 10% data usage, full fine-tuning yields at least a 4.47% gain across all prediction horizons.

However, as data availability increases (e.g., beyond 25% for 1s prediction), training from scratch eventually surpasses all fine-tuning methods. This is likely because the indoor dataset represents a task sufficiently distinct from our pre-training domain, making it easier to learn from scratch without the influence of pre-trained priors. We hypothesize that this result would no longer hold if substantial indoor data were included in the pre-training set. Similar results are also observed in the semantic occupancy forecasting task, where all VAE/CFM combinations based on the pre-trained model outperform training from scratch in mIoU. Notably, with just 10% of the semantic fine-tuning data—which is only 5% of the total data used in prior methods—full-parameter fine-tuning yields

Table 5: Pretraining performance on different LiDAR representations, without future trajectory. *: ViDAR only released weights trained on $\frac{1}{8}$ dataset.

| Methods | IoU↑ | | | |
|---|---|---|---|---|
| | 1s | 2s | 3s | Avg |
| ViDAR (I2L)* | 13.11 | 12.48 | 11.78 | 12.46 |
| Occ4D (L2L) | 26.96 | 19.51 | 16.81 | 21.09 |
| Ours-$o_v$ | **26.98** | **21.56** | **18.26** | **22.27** |
| Ours-$o_d$ | 39.64 | 29.35 | 23.73 | 30.91 |

an 82.6%/80.72%/69.70% relative mIoU gain for 1s/2s/3s prediction, surpassing OccWorld's performance [58]. After 50% data usage, the performance of from-scratch training exceeds that of the CFM-only and LoRA CFM fine-tuning methods. Nevertheless, full-parameter VAE+CFM fine-tuning consistently achieves the best performance across all data fractions and prediction horizons.

These results confirm that foundational models can capture transferable, LiDAR-based dynamic priors from unlabeled data to support downstream tasks requiring semantic interpretation. The improvements are especially notable in low-data regimes, demonstrating a reduced dependence on human-labeled samples. Overall, across all tasks, our pre-trained model achieves up to an 11.17% absolute performance gain and outperforms training from scratch in 30 out of 36 comparison points. See the Appendix for detailed numerical values corresponding to Figure 4.

However, a key question remains: why do models that fine-tune both the VAE and CFM modules consistently outperform other settings? In other words, how does fine-tuning the VAE impact the CFM's forecasting accuracy?

To further analyze, in Table 6, we compute the similarity of latent spaces under different data scales using CKA [22] and CKNNA [18]. The similarities are measured between pairwise samples w/w.o semantic information, i.e., $o'_d$ with $o'_s$ obtained from fine-tuned VAE or VAE trained only on fine-tuning data. We observe that the effectiveness of VAE fine-tuning is primarily attributed to aligning the latent space of the new domain with that of the original domain, rather than improving reconstruction accuracy. For example, when using 100% of the fine-tuning data $o'_s$, the non-pretrained VAE achieves even higher mIoU/IoU than the pretrained one, yet the final performance shows a clear gap—suggesting that preserving latent structure plays a more crucial role. Although paired data is not available for the first two adaptation scenarios, estimating latent similarity in other domains could further support our hypothesis, details are shown in Appendix.

### 4.4 Ablation study

Table 6: **VAE fine-tuning evaluation in non-semantic to semantic adaptation.**

| Pretrain | Fine-tuning | mIoU | IoU | CKA | CKNNA | Cosine |
|----------|-------------|------|-----|-----|-------|--------|
| $o'_d$ | 10% $o'_s$ | 87.97 | 75.22 | 0.739 | 0.278 | 0.907 |
| $\varnothing$ | 10% $o'_s$ | 75.81 | 72.27 | 0.653 | 0.221 | 0.183 |
| $o'_d$ | 25% $o'_s$ | 88.91 | 76.92 | 0.721 | 0.273 | 0.900 |
| $\varnothing$ | 25% $o'_s$ | 83.09 | 75.92 | 0.619 | 0.205 | 0.226 |
| $o'_d$ | 50% $o'_s$ | 89.72 | 80.09 | 0.744 | 0.275 | 0.901 |
| $\varnothing$ | 50% $o'_s$ | 90.26 | 82.03 | 0.600 | 0.194 | 0.237 |
| $o'_d$ | 100% $o'_s$ | 92.10 | 82.56 | 0.750 | 0.275 | 0.910 |
| $\varnothing$ | 100% $o'_s$ | 92.14 | 83.41 | 0.628 | 0.199 | 0.219 |

Table 7: **Ablation study of different designs in dynamic learning.** We use 3s Mean mIoU here, all experiments trained for 100 epochs.

| Method | U-Net | CFM | 3D Conv after Temporal DiT | CFG | Scale | mIoU |
|--------|-------|-----|----------------------------|-----|-------|------|
| + Baseline | ✗ | ✗ | ✗ | ✗ | 1.0 | 17.42 |
| + UNet | ✓ | ✗ | ✗ | ✗ | 1.0 | 17.77 |
| + CFM | ✓ | ✓ | ✗ | ✗ | 1.0 | 20.14 |
| + 3D Conv | ✓ | ✓ | ✓ | ✗ | 1.0 | 20.67 |
| + CFG | ✓ | ✓ | ✓ | ✓ | 1.0 | 21.05 |
| + Rescale | ✓ | ✓ | ✓ | ✓ | 10 | 23.33 |

As mentioned in Section 3.2, we now present a breakdown of the performance improvements contributed by each individual module. As shown in Table 7, we observe consistent gains from CFM, 3D convolution, and CFG components, while using classifier-free guidance gives us the most significant improvement in both IoU and mIoU. We also examine the impact of NFE on FLOPS, FPS, and accuracy. As shown in Figure 5, taking NFE=10, we get the best forecasting accuracy while maintaining a considerable level of efficiency in terms of run time and GFlops for the semantic occupancy forecasting task. For the experiment of VAE data-efficient and other details, please see the Appendix.

## 5 Conclusion

In this work, by designing a model that can be efficiently pretrained on large-scale outdoor LiDAR data, we show it can be effectively fine-tuned on diverse downstream tasks. Our approach consistently outperforms training from scratch, especially in low-data regimes. To address inefficiencies in existing models, we introduce a new VAE structure for high-ratio LiDAR compression and a conditional flow matching approach for forecasting. These components enable strong reconstruction performance and improve training and inference efficiency without sacrificing forecasting accuracy. In addition, our VAE fine-tuning strategies further boost performance. While results are encouraging, future work is needed to extend generalization to additional environments, incorporate multi-modal inputs, and integrate with planning and control systems. Overall, our findings suggest that scalable and transferable LiDAR world models are feasible and can significantly reduce reliance on annotated data in practical applications.

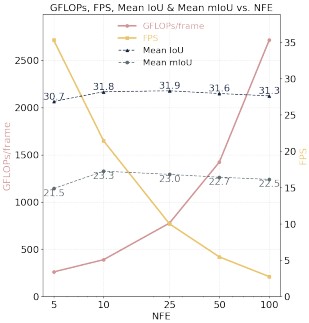

Figure 5: The performance and efficiency of proposed VAE + CFM architecture under different NFE value during the sampling process. We observe that NFE=10 corresponds to the best model performance, with reasonable efficiency in terms of FPS and Gflops per frame.

## Acknowledgments

This research was enabled in part by the Digital Research Alliance of Canada (`alliancecan.ca`).

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

# Appendix

# Contents

# A Model Setup and Further Evaluation

## A.1 Training Setup

For all flow matching based generative (foundational) model training, different from previous methods [14, 23] that used several thousand training epochs, we used 4x RTX 4090 to train models for 200 epochs with a batch size of 8. We trained the VAE part for 100 epochs with a batch size of 16 if not otherwise specified.

Following prior work[56], we also adopt the AdamW as the optimizer with $\beta_1$ and $\beta_2$ set to 0.9 and 0.99 for flow matching training, 0.99 and 0.999 for VAE training. We set the weight decay of all normalization layers to 0 and all of the other layers' to 0.001. The learning rate schedule has a linear warmup followed by cosine decay (with the minimum of the cosine decay set to be 20% of the peak learning rate). We also use EMA with a 0.9999 decay rate to ensure the updates of parameters are stable. During the training, we found that the classifier free guidance is also important in a flow-matching based framework. We randomly set 25% of historical latents in a batch to 0 during training to let the model learn to generate future latents without any conditions. In the sampling phase, we use a typical fuse method as shown in Eq. 3 to get the final output when it's the $t$-th step, where s is set to 2.

$$\hat{\mu}_\theta(\boldsymbol{z}_t, t, c; s) \;=\; (1+s)\,\boldsymbol{\mu}_\theta(\boldsymbol{z}_t, t, c) \;-\; s\,\boldsymbol{\mu}_\theta(\boldsymbol{z}_t, t, \varnothing) \tag{3}$$

Finally, we also noticed that the matching of noise scale and latent scale is important. Different from SD3's VAE, the value of latent obtained from our VAE is actually smaller, with a standard deviation of about 0.02, which makes it easy to drown the signal in noise if we use standard Gaussian noise in the training phase. Therefore, we scaled the compressed latent up by a factor of 10 in the implementation.

## A.2 Evaluation metrics

In this section, we will provide the details of all metrics used to further evaluate the performance of our proposed pipeline. All results will be presented in the last part of this section.

### A.2.1 Latent space similarity metrics

Given $\boldsymbol{o}_d \in \mathbb{R}^{H \times W \times D \times C}$ and its compressed latent $\boldsymbol{z} \in \mathbb{R}^{h \times w \times c}$, the compression ratio $\gamma$ can be calculated as in Eq. 4.

$$\gamma = \frac{h \times w \times c}{H \times W \times D \times C} \tag{4}$$

In Section 4.3, we introduce CKA (Centered Kernel Alignment) [22] and CKNNA(Centered Kernel Nearest-Neighbor Alignment )[18] to evaluate the similarity of latent spaces as both metrics are insensitive to linear transformations. Specifically, we treat each latent pixel (location) as an individual sample (i.e., every sample in CKA/CKNNA evaluation can be represented as $\boldsymbol{z}_i \in \mathbb{R}^{1 \times c}$). For the non-semantic to semantic adaptation subtask, given **k** samples from **e** latent, the representation metrics from $\boldsymbol{o}_d$ and $\boldsymbol{o}_s$ can be denoted by $\boldsymbol{X} \in \mathbb{R}^{u \times c}$ and $\boldsymbol{Y} \in \mathbb{R}^{u \times c}$ respectively. We first construct the kernel matrix:

$$\boldsymbol{K}_{ij} = \exp\left(-\frac{\|\boldsymbol{X}_i - \boldsymbol{X}_j\|^2}{2\sigma^2}\right), \quad \boldsymbol{L}_{ij} = \exp\left(-\frac{\|\boldsymbol{Y}_i - \boldsymbol{Y}_j\|^2}{2\sigma^2}\right) \tag{5}$$

Then the CKA can be calculated as:

$$\mathrm{CKA}_{\mathrm{RBF}}(\boldsymbol{X}, \boldsymbol{Y}) = \frac{\mathrm{tr}(\boldsymbol{K}_c \boldsymbol{L}_c)}{\sqrt{\mathrm{tr}(\boldsymbol{K}_c \boldsymbol{K}_c) \cdot \mathrm{tr}(\boldsymbol{L}_c \boldsymbol{L}_c)}} \tag{6}$$

where $\boldsymbol{K}_c$ and $\boldsymbol{L}_c$ are centered $\boldsymbol{K}$ and $\boldsymbol{L}$, using $\boldsymbol{K}_c = HKH$, $H = I_n - \frac{1}{n}\boldsymbol{1}_n\boldsymbol{1}_n^T$. $\boldsymbol{1}_n$ is an all-ones column vector. CKNNA is a revised version of CKA that focuses on the local manifold similarity.

For every sample $i$ in $K_c$ and $L_c$, we use cosine similarity to find out the nearest k neighbor in original samples, let $\text{knn}_{K_c}$ and $\text{knn}_{L_c}$ be the indices of its k nearest neighbors, respectively. Then we can define a mutual-KNN mask $\alpha(i, j)$ by:

$$\alpha_{ij} = \begin{cases} 1, & j \in \text{knn}_{K_c}(i) \ \wedge \ j \in \text{knn}_{L_c}(i), \\ 0, & \text{otherwise,} \end{cases} \qquad i \neq j. \qquad (7)$$

Local alignment is $\text{Align}_{KNN}(\boldsymbol{K_c}, \boldsymbol{L_c}) = \sum_{i=1}^{u} \sum_{j=1}^{u} \alpha(i, j) \, \boldsymbol{K}_c(i, j) \, \boldsymbol{L}_c(i, j)$. CKNNA is:

$$\text{CKNNA}(\mathbf{K}, \mathbf{L}) = \frac{\text{Align}_{\text{knn}}(\boldsymbol{K_c}, \boldsymbol{L_c})}{\sqrt{\text{Align}_{\text{knn}}(\boldsymbol{K_c}, \boldsymbol{K_c}) \ \text{Align}_{\text{knn}}(\boldsymbol{L_c}, \boldsymbol{L_c})}} \qquad (8)$$

Use k to represent the number of neighbors we selected, when $k \to u$, $\alpha_{ij} = 1$ for all off-diagonal pairs and CKNNA reduces to the standard CKA.

### A.2.2 Inception-based metrics

**Fidelity measurement** Following [6], we report both the 3-D Fréchet Inception Distance (FID) and the Kernel Inception Distance (KID, i.e., the squared Maximum Mean Discrepancy in feature space). Unlike FID—which assumes the Inception features form a single multivariate Gaussian and thus compares only the first two moments—KID employs a characteristic RBF kernel and provides an unbiased estimate that is sensitive to discrepancies in all higher-order statistics.

To obtain latent features from generated and ground-truth samples under comparable conditions, we retrain an autoencoder on $\boldsymbol{o}_s$. The autoencoder is based on the MinkowskiUNet32 architecture (sparse)[11], but we adapt it to our $200 \times 200 \times 16$ occupancy inputs by reducing the feature channels from $\{64, 128, 256, 512\}$ with four down-sampling stages to $\{32, 64, 128\}$ with three down-sampling stages. The training objective is to recover the compressed latent as much as possible under a cross-entropy loss.

For the output of 6 frames of future semantic occupancy, we evaluated frame-wise FID and KID. Specifically, we first extracted all non-empty voxels from the down-sampled semantic occupancy (size 25, 25, 2) and took the average, rather than performing global pooling as in previous work, to avoid the influence of empty voxels on the metrics. Given the flattened feature $\boldsymbol{z}_i^g$ and $\boldsymbol{z}_j^e$ from individual samples, we use $\boldsymbol{\mu}^g$, $\boldsymbol{\mu}^e$ and $\boldsymbol{\Sigma}^g$, $\boldsymbol{\Sigma}^e$ to represent the mean and channel covariance matrix over M ground truth samples and N estimated samples. Then we calculate FID with the following equations.

$$\text{FID} = \|\boldsymbol{\mu}^e - \boldsymbol{\mu}^g\|_2^2 + \text{Tr}\big(\boldsymbol{\Sigma}^e + \boldsymbol{\Sigma}^g - 2\big(\boldsymbol{\Sigma}^e \boldsymbol{\Sigma}^g\big)^{1/2}\big) \qquad (9)$$

For KID, we use a RBF-kernel-based unbiased-statistic estimation version as shown in equation 10.

$$\text{KID}(P, Q) = \text{MMD}_{k_\sigma}^2(P, Q)$$

$$= \frac{1}{N(N-1)} \sum_{i \neq i'} k_\sigma(\boldsymbol{z}_i^g, \boldsymbol{z}_{i'}^g) + \frac{1}{M(M-1)} \sum_{j \neq j'} k_\sigma(\boldsymbol{z}_j^e, \boldsymbol{z}_{j'}^e) - \frac{2}{NM} \sum_{i=1}^{N} \sum_{j=1}^{M} k_\sigma(\boldsymbol{z}_i^g, \boldsymbol{z}_j^e), \quad (10)$$

$$\text{where } k_\sigma(\boldsymbol{x}, \boldsymbol{y}) = \exp\Big(-\|\boldsymbol{x} - \boldsymbol{y}\|^2 / (2\sigma^2)\Big).$$

**Temporal Consistency Assessment** FID and KID provide assessments of single-frame fidelity, whilst for multi-frame continuity we adhere to previous methodology by employing FVD for evaluation. Given that no existing RGB video encoder can capture this 4D temporal information, we modified the proposed VAE to enable it to measure the continuity of 4D occupancy. Within the encoder and decoder shown in Figure 2, we respectively incorporated 3D attention/conv layers to capture temporal information.

### A.2.3 Negative log likelihood evaluation

**Computation of exact log probability density** Due to the stochastic nature of our proposed conditional flow matching model, pairwise-sample-comparison metrics such as IoU and mIoU

provide a limited measure of model quality, since these "deterministic" metrics penalize models for generating diverse but plausible predictions of the future that are substantially different from the recorded future. In other words, the model's ability to capture the uncertainty in future prediction is not measured well by the IoU and mIoU metrics. Therefore, we also evaluate the exact log probability of our CFM model that doesn't penalize it substantially for assigning probability density to other modes.

With our CFM model, $G_\theta$, we can compute the log probability of any generated future prediction or future ground-truth samples with the following methods. First, as described by previous works [10], the generative process of a continuous normalizing flow works as described below: starting from a sample from a base distribution $z_0 \in p_{z_0}(z_0)$ and a parametrized ODE $G_\theta = G(z(t), t; \theta)$ which is the flow function, we can obtain $z(t_1)$ from the target distribution by solving for the initial value problem $z(t_0) = z_0$, $\frac{\partial z(t)}{\partial t} = G$. In this process, the rate of change of log-probability density follows the instantaneous change of variables formula [10]:

$$\frac{\partial \log p(z_t)}{\partial t} = -Tr(\frac{\partial G}{\partial z(t)}) \tag{11}$$

With this equation, the total change in log probability density from $t_0$ to $t_1$ is calculated by integrating with respect to time:

$$\log p(z(t_1)) = \log p(z(t_0)) - \int_{t_0}^{t_1} Tr(\frac{\partial G}{\partial z(t)}) \, dt \tag{12}$$

Given any sample x in the target distribution, we can compute $z_0$ that generates x and the log likelihood of x by solving for the following IVP [10]:

$$\begin{cases} z_0 = \int_{t_1}^{t_0} G(z(t), t; \theta) \, dt \\ \log p(x) - \log p_{z_0}(z_0) = \int_{t_1}^{t_0} -Tr(\frac{\partial G}{\partial z(t)}) \, dt \end{cases} \tag{13}$$

with $z(t_1) = x$ and $\log p(x) - \log p(t_1) = 0$.

After obtaining the change in log probability density, we can add this change to the log probability of the prior distribution to determine the exact log likelihood of x.

Practically, as what is done for FFJORD [13], the trace of the Jacobian of the flow function can be approximated with the Hutchinson's trace estimator which takes $o(n)$. Moreover, we use a standard Gaussian as the base distribution which gives:

$$\log p_{z_0}(z_0) = -\frac{1}{2} \left( \|z_0\|^2 + D \log(2\pi) \right) \tag{14}$$

where D is the number of scalar dimensions of the sample. This enables us to determine the exact log likelihood of any generated sample or sample from the future occupancy latent space (any sample from the future occupancy distribution) numerically.

**Details on NLL evaluation**   For all log likelihood values computed, we use an Euler ODE solver with a step size of 0.02, relative and absolute tolerance of $1 \times 10^{-5}$. The trace of the Jacobian is obtained by utilizing Hutchinson's trace estimator with the probe vector sampled from a Rademacher distribution.

After obtaining the exact log probability, we evaluate the negative log likelihood in the form of bits-per-dimension (BPD) which is obtained by:

$$BPD(x) = -\frac{\log p(x)}{D \ln 2} \tag{15}$$

where D is the number of scalar dimensions of the latent sample x and the division by $\ln 2$ converts the unit to bits.

**DDPM log likelihood computation**   For discrete DDPM-based model, we need to construct and solve the IVP introduced previously with a mathematically equivalent representation for the flow function $G$.

For a model (i.e. DOME [14] which is constructed based on a 1000-step DDPM) whose noise variances $\{\beta_k\}_{k=0}^{N-1}$ are linearly spaced, to reuse the same parameters for likelihood evaluation, we can embed this chain in the piece-wise–constant *variance–preserving* SDE

$$d\mathbf{x}_t = -\frac{1}{2}\,\beta(t)\,\boldsymbol{x}_t\,dt + \sqrt{\beta(t)}\,d\boldsymbol{w}_t, \quad \beta(t) = \beta_k \quad \text{for} \quad t \in \left[\frac{k}{N-1}, \frac{k+1}{N-1}\right). \tag{16}$$

We can write its deterministic form [43]

$$\dot{\boldsymbol{x}}_t = G(\boldsymbol{x}_t, t; \theta) = -\frac{1}{2}\,\beta(t)\,\boldsymbol{x}_t - \frac{1}{2}\,\beta(t)\,\boldsymbol{s}_\theta(\boldsymbol{x}_t, t). \tag{17}$$

with score $\boldsymbol{s}_\theta(\boldsymbol{x}_t, t) = -\hat{\varepsilon}_\theta(\boldsymbol{x}_t, t)/\sigma(t)$. With this approximation of the flow function, the IVP can be constructed and solved following the same procedure as the CFM model.

### A.3 More results on semantic occupancy forecasting task

The NLL values in terms of BPD and the corresponding standard deviation (across the entire validation set; we use ground-truth samples not the generated predictions) obtained from the semantic occupancy forecasting task are shown in table 8. Our semantic occupancy forecasting model conditioned on historical/past trajectory achieves comparable NLL values with DOME [14]: only 0.25% absolute difference (4.1% relative) while DOME utilizes future trajectory in the condition. For our model conditioned on future trajectory, the NLL value is lower than that of DOME by 25% relative difference (1.49% absolute). This shows that our CFM model better fits the true future-occupancy distribution than the previous (stochastic) SOTA model.

In terms of standard deviation in NLL, for our models (conditioned on either history or future trajectory), the standard deviation is at least 86% (relative) less than that of DOME. This statistical metric further supports the conclusion drawn above by demonstrating the high consistency of our model's performance in terms of assigning high probability to correct futures.

Table 8: Mean negative log-likelihood for different semantic occupancy forecasting models.

| Model | Mean NLL (bits/dim) | Std. NLL |
|---|---|---|
| Ours (Hist. traj.) | 6.29 | **0.04** |
| DOME (Fut. traj.) | 6.04 | 0.28 |
| Ours (Fut. traj.) | **4.55** | **0.02** |

Different from image, LiDAR occupancy from a BEV perspective has a clear depth correspondence between grid cells and environments. However, the global pooling operation will take the average of features in all grid cells, agnostic to geometric location. Here, inspired by [39], we also measured the FID and MMD of features from different depth bins. Specifically, in the obtained $\boldsymbol{z}_i^e \in \mathbb{R}^{25 \times 25 \times 2}$, every cell stands for $3.2m \times 3.2m$ area in the environments. Based on the distance from the center point, we divide the area into three zones: within ±8 m, ±8 to ±24 m, and ±24 to ±40 m. We then take the average of the non-empty voxels in each zone and concatenate them to get the final feature for evaluation, denoted as $\text{FID}_r$ and $\text{KID}_r$ in Table 9.

Table 9: $\text{FID}_r$ and $\text{KID}_r(\times 10^{-2})$ comparison for our method and other generative-based and deterministic-based models.

| Method | 1s | | 2s | | 3s | | AVG | |
|---|---|---|---|---|---|---|---|---|
| | $\text{FID}_r$ | $\text{KID}_r$ | $\text{FID}_r$ | $\text{KID}_r$ | $\text{FID}_r$ | $\text{KID}_r$ | $\text{FID}_r$ | $\text{KID}_r$ |
| OccWorld | 46.20 | 6.98 | 44.91 | 6.51 | 44.00 | 6.23 | 45.03 | 6.57 |
| DOME | 18.09 | 2.28 | 21.56 | 2.87 | 25.03 | 3.46 | 21.56 | 2.87 |
| Ours (Hist. traj) | 7.20 | 0.78 | 12.39 | 1.47 | 17.80 | 2.49 | 12.46 | 1.58 |
| Ours (Fut. traj) | 6.71 | 0.74 | 12.29 | 1.54 | 16.82 | 2.32 | 11.94 | 1.53 |

Our model establishes a new state of the art on both FID and KID. On average, for our model without future trajectories, it reduces FID to just 28.3%/48.5% and KID to 22.7%/44.4% of the

scores reported by the deterministic (OccWorld) and stochastic (DOME) models, respectively. The advantage remains under the stricter metrics $FID_r$ and $KID_r$: our method attains only 23.36%/48.79% ($FID_r$) and 19.30%/44.25% ($KID_r$) of the corresponding values.

Interestingly, we observed that the $FID_r/KID_r$ of OccWorld decreases slightly (around 2%) when we forecast longer-term future results. We attribute this to the auto-regressive displacement forecasting design in OccWorld, which preserves more diversity at longer horizon, although the overall performance is lower. Specifically, we calculated the category variance at different time horizons (all numbers follow in $\times 10^{-3}$): for the ground truth, the variance fluctuated between 3.055 and 3.021, while in Occworld, it rose from 2.865 at t=0.5 seconds to 3.038, and our model fell from 3.086 to 2.821.

## B  Dataset introduction

As mentioned earlier, we used nuScenes Semantic Occupancy ($o_s$), KITTI360 raw sweeps[25], and our own collection of indoor LiDAR data to test the three subtasks. In the semantic occupancy forecasting task, we further divide nuScenes semantic occupancy data into pretrain set ($o_s'$) and fine-tuning set. For pretraining, we use voxelized LiDAR sweeps in the nuScenes dataset. In Table 10, we list the specific number of frames of all these datasets after down-sampling to 2Hz.

Table 10: Training-validation set splitting for each dataset

| Dataset | training set | validation set |
|---|---|---|
| $o_s$ | 19728 | 4219 |
| $o_s'$ | 11208 | 4219 |
| KITTI-360 | 12093 | 4912 |
| Indoor LiDAR Dataset | 23504 | 9720 |
| nuScenes LiDAR sweep | 22632 | 4368 |

Specifically, for the indoor LiDAR dataset, it consists of 25 sequences that we collected in 2 different scenarios, using an OSO-128 LiDAR sensor with a 90-degrees vertical field of view. A robot, modeled with unicycle kinematics, was manually driven through a university building over the course of five days. An expert human operator controlled the robot, skillfully avoiding pedestrians while navigating toward designated goals. As shown in Figure 6, compared to the data in the autonomous driving scenario, where the vast majority of vehicles move parallel to ego vehicles, the data we collected includes more complex trajectories with more objects (people).

As we mentioned in the main text, the amount of data after downsampling is still large considering that we only collected this data in 2 scenes. This makes it easier for the model to learn domain-specific knowledge and explains why the performance of our pretrained model drops after using more than 50% of the data.

## C  Detailed pretraining and fine-tuning results

In this section, we explain more details about the results from our experiments on how different pretrained/from-scratch VAE and CFM combinations perform on each of the three subtasks, which, again, are high-beam LiDAR adaptation, indoor occupancy forecasting, and semantic occupancy forecasting. For LoRA[17], across all our downstream tasks, we choose a rank of 128 and $\alpha$ of 128. Please check section E for details on this selection. All models present in this section are trained for 40 epochs.

### C.1  High-beam adaptation and indoor occupancy forecasting

In Table 11 and Table 12, we organize the performance of each of the pretrain/from-scratch combinations (VAE pretrained and CFM pretrained and fine-tuned with LoRA[17], both pretrained and full-parameter fine-tuned, only CFM pretrained, and both from-scratch) for high-beam adaptation and indoor occupancy forecasting in terms of IoU. IoU is computed through inferring the trained CFM

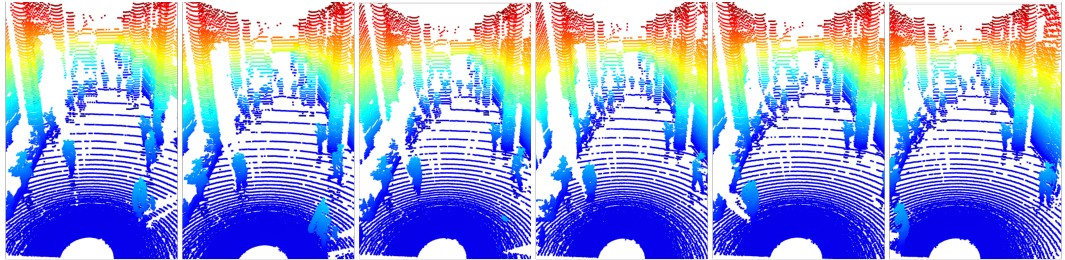

(a) An enclosed indoor corridor with pedestrians walking parallel to the robot's direction of motion.

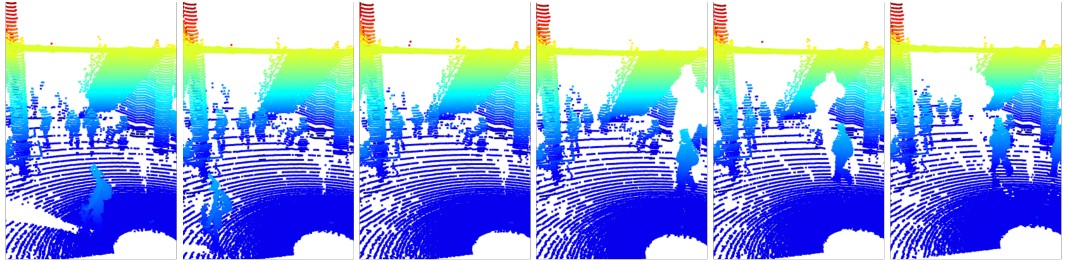

(b) An indoor intersection with complex pedestrian trajectories.

Figure 6: Indoor dataset: We collected over 200K raw point cloud frames (was down-sampled to around 23000 frames in training) at two indoor locations, incorporating complex movements of the crowd and changes in the indoor scenery.

model with NFE=10. For the high-beam adaptation task, the pretrained CFM model is trained on original LiDAR sweeps, denoted by $G_\theta^v$. From the result, it is evident that full-parameter pretrained (both or only-CFM, with or without LoRA) yields exceeding performance over from-scratch trainings. Specifically, at 10% of fine-tuning data usage, our pretrained model with full parameter fine-tuning achieves more than 14.48% relative performance improvement (more than 1.5% absolute performance improvement) across the three-second forecasting horizon and even at 100% data usage, our model still offers an absolute performance growth of about 1%. This demonstrates the effectiveness of our pretrained foundational models in transferring geometric knowledge learned from 32 beam LiDAR data to tasks that demand different geometric properties (64 beams in this case) at different availability of fine-tuning data.

For the indoor occupancy forecasting task, we use the same pretrained CFM model as the high-beam adaptation task. From the result, it is shown that at low data availability for fine-tuning (10% and 25%), our pretrained models achieve exceeding performance over the from-scratch training. But when the amount of fine-tuning data increases to more than 50%, the from-scratch training obtains comparable or even better performance. Again, we explain this performance threshold to be resulting from the fine-tuning data amount which overrides the pretraining data and the lack of variance in the geometry of the indoor occupancy dataset collected.

### C.2 Semantic Occupancy Forecasting

For semantic occupancy forecasting task, the pretrained CFM model is trained on densified LiDAR frames, denoted by $G_\theta^d$. The results with numerical value of mIoU and IoU for different pretrained/from-scratch VAE and CFM combinations (NFE=10) at different fine-tuning data fractions are presented in Table 13. It is shown that at 10% of fine-tuning data utilization, our pretrained VAE and CFM with full-parameter fine-tuning provides a 11.17% absolute forecasting performance improvement in mIoU. In general, even the performance gain decays as the amount of fine-tuning data usage increases (which makes sense as it is getting closer and closer to the amount of pretraining data), the effectiveness of our foundational model prevails across all fine-tuning data fractions (at 100%, we still have about 1% performance improvement over the from-scratch model). One key thing to note is that we only fine-tune/train-from-scratch our semantic occupancy forecasting model with half of the available semantic labels for nuScenes (50% of the data used for the same task for other baseline models). This additional splitting of the training set is to prevent the overlap between pretrain and fine-tuning data in terms of geometric structure, as both densified occupancy forecasting

Table 11: High-beam adaptation fine-tuning results.

| Data Fraction | Horizon | LoRA | Full-parameter | Only CFM-pretrained | Train from scratch |
|---|---|---|---|---|---|
| 10% | 1s | 18.57 | 20.69 | 19.23 | 16.42 |
| | 2s | 14.03 | 15.11 | 14.09 | 12.27 |
| | 3s | 11.77 | 12.33 | 11.49 | 10.77 |
| 25% | 1s | 22.70 | 21.74 | 20.45 | 18.53 |
| | 2s | 16.68 | 16.23 | 14.97 | 13.92 |
| | 3s | 13.77 | 13.03 | 12.00 | 11.69 |
| 50% | 1s | 23.70 | 22.78 | 21.95 | 21.43 |
| | 2s | 17.70 | 17.17 | 16.21 | 15.84 |
| | 3s | 14.60 | 14.02 | 13.17 | 13.01 |
| 100% | 1s | 24.16 | 24.50 | 24.00 | 23.40 |
| | 2s | 18.50 | 18.28 | 17.66 | 17.32 |
| | 3s | 15.39 | 14.93 | 14.16 | 14.08 |

Table 12: Indoor occupancy forecasting fine-tuning results.

| Data Fraction | Horizon | LoRA | Full-parameter | Only CFM-pretrained | Train from scratch |
|---|---|---|---|---|---|
| 10% | 1s | 29.40 | 30.40 | 29.73 | 27.28 |
| | 2s | 26.60 | 26.45 | 25.82 | 24.96 |
| | 3s | 25.41 | 24.99 | 24.23 | 23.92 |
| 25% | 1s | 33.06 | 33.38 | 32.90 | 32.34 |
| | 2s | 28.70 | 29.36 | 28.70 | 29.10 |
| | 3s | 26.93 | 27.55 | 26.70 | 27.44 |
| 50% | 1s | 33.58 | 34.27 | 34.08 | 35.13 |
| | 2s | 28.97 | 30.19 | 29.73 | 31.24 |
| | 3s | 27.11 | 28.21 | 27.71 | 29.17 |
| 100% | 1s | 34.49 | 35.64 | 35.32 | 36.52 |
| | 2s | 30.11 | 31.71 | 31.07 | 32.50 |
| | 3s | 28.18 | 29.60 | 28.98 | 30.48 |

and semantic occupancy forecasting models share the same set of scenes and the difference is that semantic occupancy forecasting data is labeled by category. This means that at 10% of fine-tuning data usage, having both VAE and CFM full-parameter fine-tuned based on our foundational model is able to achieve comparable performance with training-from-scratch and OccWorld ($<\pm1.5\%$ absolute performance difference in both IoU and mIoU across the three-second forecasting horizon) with only 5% of the data it consumes. Please refer to section H for visualizations of the forecasting results.

Table 13: Breakdown of fine-tuning results on the semantic occupancy task.

| Data Fraction | Forecasting Horizon | LoRA | | Full-parameter | | Only CFM Pretrain | | From Scratch | |
|---|---|---|---|---|---|---|---|---|---|
| | | IoU | mIoU | IoU | mIoU | IoU | mIoU | IoU | mIoU |
| 10% | 1s | 35.92 | 23.29 | 36.24 | 25.95 | 36.05 | 24.09 | 27.00 | 13.52 |
| | 2s | 26.01 | 14.19 | 26.17 | 14.91 | 25.61 | 14.63 | 19.41 | 8.25 |
| | 3s | 20.60 | 9.80 | 20.87 | 11.09 | 20.24 | 10.21 | 15.81 | 6.14 |
| 25% | 1s | 37.21 | 27.42 | 37.85 | 28.88 | 36.64 | 27.80 | 33.84 | 22.73 |
| | 2s | 27.46 | 17.23 | 27.09 | 17.58 | 26.95 | 17.03 | 24.62 | 13.65 |
| | 3s | 22.04 | 12.31 | 21.36 | 12.30 | 21.61 | 11.84 | 19.29 | 9.38 |
| 50% | 1s | 37.17 | 28.47 | 38.40 | 30.84 | 37.09 | 28.23 | 37.02 | 28.10 |
| | 2s | 27.59 | 17.48 | 27.81 | 19.00 | 27.53 | 17.64 | 27.30 | 17.50 |
| | 3s | 22.31 | 12.44 | 22.14 | 13.62 | 22.29 | 12.58 | 21.80 | 12.34 |
| 100% | 1s | 37.13 | 28.83 | 38.91 | 31.64 | 36.59 | 28.39 | 38.21 | 30.54 |
| | 2s | 27.80 | 17.56 | 28.58 | 19.59 | 27.46 | 18.01 | 28.22 | 19.06 |
| | 3s | 22.65 | 12.52 | 23.13 | 14.22 | 22.50 | 13.02 | 22.31 | 13.45 |

# D   More details on representation alignment

As we mentioned before, during the pretraining phase, we found that it is more appropriate to use new data in fine-tuning the original VAE as opposed to training a VAE from scratch, even though the performance of our vae trained from scratch was excellent. We attribute this to the fact that the pretrained generative model is in fact more adapted to the data distribution of the original latent, whereas the subspace structure of a latent learned from scratch with fine-tuned data would in fact be very different.

Specifically, for sub-task 3 (nonsemantic-to-semantic transfer), we cannot directly fine-tune the VAE considering the inconsistent size of the embedding layers. Instead, we use the approach in Fig. 7: considering that $o_s$ and $o_d$ match one-to-one, and the original VAE is fine-tuned on $o_d$, we can directly constrain the direct outputs of the encoder: mu and sigma. Previous work has tried to use bidirectional KL or EMD distance as a loss, but this does not work in our case: the data for the two modalities are already very close, differing only by semantic information.

Table 14: Comparison of feature alignment metrics before and after alignment.

|  | Dual KL | EMD | CKA | CKNNA | CosSim |
|---|---|---|---|---|---|
| **Dense-Sem feature** | 0.004 | 0.008 | 0.739 | 0.278 | 0.183 |
| **Dense-Sem (Aligned) feature** | 0.002 | 0.004 | 0.944 | 0.410 | 0.907 |

Therefore we compared the latent from pair-wise $o_d$ and $o_s$ in line 1 of Table 14. We can see that both Dual-KL and EMD are very close, only cosine similarity is very different, which actually show us that the most significant difference in the latent learned from these two VAEs is in fact the direction of the high-dimensional tensors. Then after using cosine similarity to constrain the $o_s$ VAE learning, the result is shown in line 2. With the tensor orientation restriction, we can successfully align the latent of the two VAEs, and the subsequent generative model training demonstrates that our alignment scheme can further improve the results.

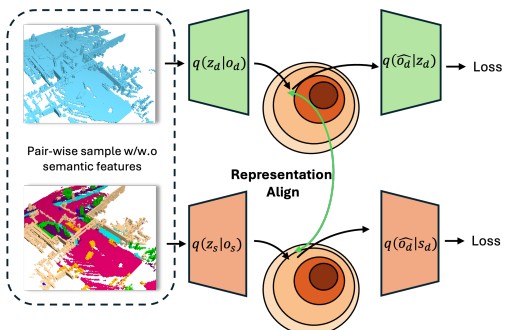

Figure 7: Use pretrained VAE for $o_d$ to implicitly guide the structure of $o_s$ latent space

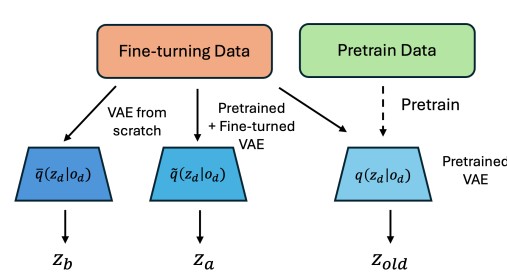

Figure 8: The comparison of latent space structure for non-paired data (in subtask 1 and 2)

For different beam LiDAR adaptation and outdoor-indoor generalization, while we can directly fine-tune the VAE, we cannot directly compare the similarity of the subspaces (CKA[22] and CKNNA[18] require the same or pair-wise samples as inputs to different networks). Our solution is shown in Figure 8. First, we use the original pretrained VAE, denoted by $q(z_d|o_d)$ to get the latent of fine-tuning data without any training (adaptation), denoted by $z_{old}$. Then we use the fine-tuned and training from scratch VAE to obtain the $z_b$ and $z_a$. If our hypothesis is correct, the distance between $(z_a, z_{old})$ should be smaller than $(z_b, z_{old})$.

As data shown in Table 15 and 16, we listed the $(z_a, z_{old})$ and $(z_b, z_{old})$ in every 2 continuous lines for different fractions of fine-tuning data. We observe that with the increase of the data amount used to fine-tuning the VAE, we can have better results in terms of IoU, even better (when using 100% high beam KITTI data) or very close to the pretrained one. However, what really determines the forecast performance is actually the similarity between the latent space after fine-tuning the VAE and the latent space after only pretraining. Compared to training from scratch, the latent space after

fine-tuning the VAE is more similar to the latent produced by using pretrained weight directly, which allows for better use of the knowledge from the foundational model. In other words, fine-tuning the VAE is in fact about leaving the original latent space as unaltered as possible while allowing the weights of the original VAE to be adapted (and accurately compressed) to the new sample.

Table 15: **VAE fine-tuning evaluation in different beam adaptation**

| Pretrain | Fine-tuning | IoU | CKA | CKNNA |
|---|---|---|---|---|
| $o_v$ | 10% KITTI | 97.97 | 0.661 | 0.231 |
| $\varnothing$ | 10% KITTI | 85.08 | 0.477 | 0.171 |
| $o_v$ | 25% KITTI | 98.16 | 0.625 | 0.214 |
| $\varnothing$ | 25% KITTI | 88.66 | 0.489 | 0.188 |
| $o_v$ | 50% KITTI | 98.31 | 0.610 | 0.211 |
| $\varnothing$ | 50% KITTI | 90.79 | 0.495 | 0.188 |
| $o_v$ | 100% KITTI | 98.40 | 0.611 | 0.208 |
| $\varnothing$ | 100% KITTI | 98.46 | 0.494 | 0.189 |

Table 16: **VAE fine-tuning evaluation in outdoor to indoor adaptation.**

| Pretrain | Fine-tuning | IoU | CKA | CKNNA |
|---|---|---|---|---|
| $o_d$ | 10% Indoor | 95.82 | 0.739 | 0.245 |
| $\varnothing$ | 10% Indoor | 88.77 | 0.592 | 0.186 |
| $o_d$ | 25% Indoor | 96.65 | 0.720 | 0.233 |
| $\varnothing$ | 25% Indoor | 92.14 | 0.606 | 0.199 |
| $o_d$ | 50% Indoor | 97.15 | 0.691 | 0.221 |
| $\varnothing$ | 50% Indoor | 94.09 | 0.611 | 0.203 |
| $o_d$ | 100% Indoor | 98.18 | 0.688 | 0.220 |
| $\varnothing$ | 100% Indoor | 95.97 | 0.609 | 0.200 |

# E   Analysis on LoRA fine-tuning with different ranks

To determine the optimal LoRA[17] configuration for our downstream tasks, we tested the following ranks: 32, 64, and 128 with $\alpha = rank$ and a dropout rate of 0.05 on the semantic occupancy forecasting subtask with 10% of fine-tuning data and both VAE and CFM components pretrained as explained earlier. We apply LoRA only to the linear layers, embedding layers, 2D and 3D convolution layers in the CFM architecture. The training is done for 40 epochs for LoRA and full-parameter fine-tuning baseline. The goal of this mini-scale experiment is to find the balance between model performance and parameter efficiency. The result is shown in table 17. First, after sampling the CFM, the model yields suboptimal performance under the rank of 32 and 64, with 3.51% and 2.48% absolute performance gaps with the full-parameter fine-tuning in terms of mean mIoU across three-second forecasting horizon, respectively. This indicates that the model does not acquire a satisfactory level of semantic information from fine-tuning data within the 40 epochs training process which might be caused by lack of depth in the LoRA layers. On the other hand, rank of 128 demonstrates performance that is closest to the full-parameter fine-tuning: 0.25% less in mean IoU and 0.91% less in mean mIoU. This shows that a rank of 128 offers enough depth for information to be extracted from the semantic occupancy data and demonstrates comparable performance against the baseline. This rank and $\alpha$ combination is what we presented earlier for all downstream tasks under the "LoRA" fine-tuning category.

We did not test any higher or lower rank because: since both ranks of 32 and 64 have a considerable performance gap compared to the full-parameter fine-tuning, lower rank means even less parameters (depth) for LoRA layers which makes the model less likely to learn useful information from the fine-tuning data and consequently, is unlikely to achieve comparable performance with the full-parameter method; on the other hand, a rank of 128 already has 10.21 million trainable parameters in total where our CFM model alone only has 11.51 million parameters. This means that under such rank, we are fine-tuning about the same number of parameters as the full-parameter fine-tuning approach.

Table 17: Comparison of different LoRA configurations (rank and $\alpha$ values) and full parameter fine-tuning for semantic occupancy forecasting task. We report the method type, number of trainable parameters, mean IoU, and mean mIoU. Note that our full-parameter CFM models only have 11.51 M parameters, excluding VAE which has 18.86 M parameters.

| Method | Rank | $\alpha$ | Parameters (M) | mean IoU $\uparrow$ | mean mIoU $\uparrow$ |
|---|---|---|---|---|---|
| Full-parameter | - | - | 11.51 | 27.76 | 16.67 |
| LoRA | 32 | 32 | 2.55 | 26.27 | 13.16 |
| LoRA | 64 | 64 | 5.10 | 27.14 | 14.19 |
| LoRA | 128 | 128 | 10.21 | 27.51 | 15.76 |

# F  Additional ablation studies

## F.1  The data efficiency of proposed VAE structure

Follow the official split of train/validation set in nuScenes, we test the proposed VAE performance under different fractions of training data. As shown in Table 18, our method only needs half of the training data to exceed all of the previous data compressors, as we mentioned in the abstract.

Table 18: Performance of proposed VAE under different training data fraction, under 192x compression rate, 100 epochs training

| Data fraction | IoU | mIoU |
|---|---|---|
| 100% | 85.8 | 93.8 |
| 50% | 83.4 | 92.1 |
| 25% | 82.0 | 90.2 |
| 10% | 78.9 | 85.8 |

## F.2  VAE and forecasting result break down

Table 19: Per-class IoU and overall metrics across different semantic occupancy compression models.

| Method | Comp. | mIoU | IoU | Others | Barrier | Bicycle | Bus | Car | Const. Veh. | Motorcycle | Pedestrian | Traffic Cone | Trailer | Truck | Drive. Surf. | Other Flat | Sidewalk | Terrain | Man-made | Vegetation |
|---|---|---|---|---|---|---|---|---|---|---|---|---|---|---|---|---|---|---|---|---|
| OccWorld[58] | 16 | 65.7 | 62.2 | 45.0 | 72.2 | 69.6 | 68.2 | 69.4 | 44.4 | 70.7 | 74.8 | 67.6 | 56.1 | 65.4 | 82.7 | 78.4 | 69.7 | 66.4 | 52.8 | 43.7 |
| OccSora[45] | **512** | 27.4 | 37.0 | 11.7 | 22.6 | 0.0 | 34.6 | 29.0 | 16.6 | 8.7 | 11.5 | 3.5 | 20.1 | 29.0 | 61.3 | 38.7 | 36.5 | 31.1 | 12.0 | 18.4 |
| OccLLAMA[48] | 16 | 75.2 | 63.8 | 65.0 | 87.4 | 93.5 | 77.3 | 75.1 | 60.8 | 90.7 | 88.6 | 91.6 | 67.3 | 73.3 | 81.1 | 88.9 | 74.7 | 71.9 | 48.8 | 42.4 |
| DOME[14] | 64 | 83.1 | 77.3 | 36.6 | 90.9 | 95.9 | 85.8 | 92.0 | 69.1 | 95.3 | 96.8 | 92.5 | 77.5 | 85.6 | 93.6 | 94.2 | 89.0 | 85.5 | 72.2 | 58.7 |
| Ours | 192 | **92.8** | **85.8** | **88.5** | **97.8** | **97.5** | **93.7** | **96.0** | **86.2** | **98.4** | **97.6** | **97.6** | **92.1** | **94.7** | **97.2** | **98.5** | **95.8** | **94.8** | **83.6** | **68.3** |

In Table 19, we present the compression performance of the proposed VAE across different semantic categories. With a 3× compression ratio, our method consistently outperforms prior approaches in all categories. Notably, for rare classes such as Construction Vehicle, Trailer, and Vegetation, our model achieves improvements of 22.2%, 18.8%, and 25.4% over DOME's VAE, respectively. These results indicate that our VAE effectively mitigates the impact of data imbalance across categories in the compression process.

Table 20: 3 seconds average per-class IoU for semantic occupancy forecasting

| Method | mIoU | IoU | Others | Barrier | Bicycle | Bus | Car | Const. Veh. | Motorcycle | Pedestrian | Traffic Cone | Trailer | Truck | Drive. Surf. | Other Flat | Sidewalk | Terrain | Man-made | Vegetation |
|---|---|---|---|---|---|---|---|---|---|---|---|---|---|---|---|---|---|---|---|
| OccWorld[58] | 17.13 | 26.63 | 12.23 | 20.77 | 8.27 | 20.50 | 19.86 | 12.58 | 7.89 | 8.95 | 8.45 | 13.04 | 17.73 | 35.09 | 23.65 | 23.97 | 20.66 | 17.01 | 20.17 |
| Ours (Hist. Traj.) | **23.33** | **31.78** | **21.41** | **23.70** | **15.33** | **26.00** | **23.05** | **27.44** | **13.19** | **10.29** | **13.10** | **21.89** | **24.85** | **40.68** | **30.99** | **29.29** | **26.64** | **21.52** | **26.57** |
| DOME[14] | 22.18 | 32.13 | 19.84 | 25.66 | 15.36 | 21.03 | 21.98 | 23.96 | 11.36 | 7.99 | 14.79 | 18.02 | 21.58 | 39.84 | 30.46 | 28.74 | 25.35 | 23.01 | 27.22 |
| Ours (Fut. Traj.) | **28.49** | **37.52** | **28.38** | **33.77** | **19.99** | **29.81** | **28.17** | **32.46** | **18.76** | **12.08** | **20.78** | **25.59** | **30.85** | **43.27** | **34.01** | **32.65** | **29.34** | **30.10** | **34.26** |

Now, we present a per-category performance analysis. In Table 20, we compare the average IoU for 3s under each category. Compared to OccWorld[58] , our improvement in forecasting performance for medium-sized objects is particularly significant: we improve the forecasts of bicycle, Const. Veh. and Motorcycle by 85.36%, 118.12% and 67.17%, respectively. A broad performance gain is also observed when including future trajectory as a condition, the top 2 improvement categories are also small objects, relatively 65.18% and 51.2% for Motorcycle and Pedestrian respectively.

## F.3  Forecasting performance as a function of the VAE's compression ratio

We present the reconstruction rates of the VAE architecture at various compression ratios in Table 2. Then a further question arises: which category of information is more susceptible to loss? In Table

21, we show the performance drop for 4 listed reconstruction rates. Interestingly, we notice that the

Table 21: Proposed VAE per category reconstruction rate break down for different compression ratio

|  | Others | Barrier | Bicycle | Bus | Car | Const. Veh. | Motorcycle | Pedestrian | Traffic Cone | Trailer | Truck | Drive. Surf. | Other Flat | Sidewalk | Terrain | Man-made | Vegetation |
|---|---|---|---|---|---|---|---|---|---|---|---|---|---|---|---|---|---|
| **x32** | 98.56 | 99.38 | 98.61 | 99.54 | 99.68 | 97.69 | 99.59 | 99.46 | 99.07 | 98.84 | 99.21 | 99.71 | 99.59 | 99.21 | 98.83 | 97.87 | 93.66 |
| **x192** | 88.51 | 97.80 | 97.45 | 93.63 | 96.00 | 84.19 | 98.42 | 97.65 | 97.68 | 89.12 | 92.71 | 97.27 | 98.56 | 94.40 | 93.86 | 83.66 | 68.27 |
| **x384** | 79.56 | 95.14 | 96.03 | 87.75 | 88.94 | 75.26 | 96.34 | 95.28 | 95.33 | 81.89 | 85.67 | 93.01 | 96.61 | 86.94 | 86.61 | 72.52 | 58.32 |
| **x768** | 66.14 | 88.74 | 90.77 | 79.57 | 78.89 | 62.37 | 90.49 | 88.07 | 87.73 | 69.97 | 76.63 | 87.55 | 90.74 | 77.29 | 76.03 | 62.53 | 50.56 |
| **Drop (x32->x192)** | 10.05 | 1.58 | 1.16 | 5.91 | 3.68 | 13.50 | 1.17 | 1.81 | 1.39 | 9.72 | 6.50 | 2.44 | 1.03 | 4.81 | 4.97 | 14.21 | 25.39 |
| **Drop (x32->x384)** | 19.00 | 4.24 | 2.58 | 11.79 | 10.74 | 22.43 | 3.25 | 4.18 | 3.74 | 16.95 | 13.54 | 6.70 | 2.98 | 12.27 | 12.22 | 25.35 | 35.34 |
| **Drop (x32->x768)** | 32.42 | 10.64 | 7.84 | 19.97 | 20.79 | **35.32** | 9.10 | 11.39 | 11.34 | 28.87 | 22.58 | 12.16 | 8.85 | 21.92 | 22.80 | **35.34** | **43.10** |

voxel missing in the small and rare foreground objects is much less severe than in the background (vegetation, manmade) and large objects (construction vehicle) during compression ratio increase, while these small targets are more important for safe driving.

Next we show the forecast performance as a function of VAE for different compression rates. First, the results of CFM based on latent with different compression ratios are shown below. For time considerations, all CFMs here are trained for 200 epochs. At the highest reconstruction rate of x32,

Table 22: Forecasting result with different compression ratio VAE

| Compression | 1 s IoU | 1 s mIoU | 2 s IoU | 2 s mIoU | 3 s IoU | 3 s mIoU | Avg IoU | Avg mIoU |
|---|---|---|---|---|---|---|---|---|
| x32 | 38.40 | 28.96 | 28.54 | 18.52 | 23.14 | 13.48 | 30.02 | 20.31 |
| x192 | **40.53** | **33.17** | **30.37** | **21.09** | **24.44** | **15.64** | **31.78** | **23.33** |
| x384 | 39.68 | 32.96 | 29.95 | 20.91 | 24.25 | 14.98 | 31.29 | 22.95 |
| x768 | 34.66 | 27.10 | 24.51 | 17.30 | 22.61 | 12.95 | 27.26 | 19.11 |

we find that the network is not fully fitted at 200 epochs, and a longer training time may allow the network to fit better. However, since this is clearly contrary to the 'sample/training efficiency' of this paper, we keep the training time the same for fair evaluation. For latents with higher compression multiples, the forecast performance continues to degrade. We believe that this is not only due to the fact that it retains less information on large objects, but also because at larger compression multiples, latents become very sensitive to noise, i.e., the latent space loses smoothness. We find that at multiples of x384 and x768, noise of the order of 1e-4 can crash the network (the magnitude of the latent quantity is approximately between 1e-2 and 1e-3).

## F.4 Model performance with different fractions of pretraining data

In the previous section, we showed that based on the foundational model, the performance boost when we increase the fine-tuning data amount and also the result w/w.o the foundational model. But another question is whether the performance of the model improves with the inclusion of more pretraining data? We designed a simple experiment to answer this question.

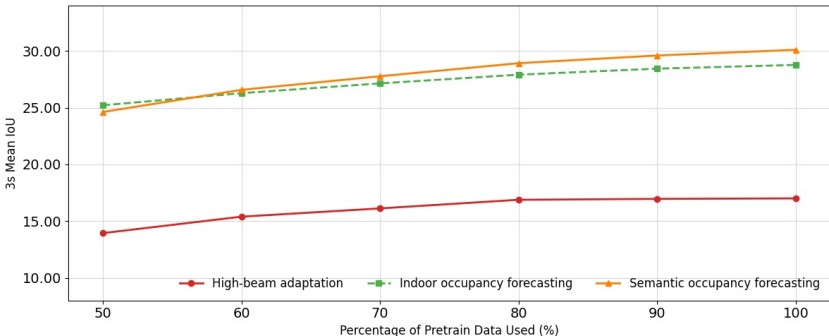

Figure 9: Performance improvement from different ratios of pretraining data

We choose 25% of fine-tuning data for 3 different subtasks to simplify the fine-tuning. In Fig. 9, 100% data on the x-axis represents the full pretraining data $O_v$. We observe a clear performance

improvement (relative improvements of 21.9% , 22.2%, and 12.9% for indoor occupancy forecasting, semantic occupancy prediction, and high-beam adaptation tasks) as the amount of pretraining data varies from 50% of $O_v$ to 100%.

### F.5 CFM performance sensitivity analysis with sampling randomization

The inference process of flow matching is essentially solving the Probability Flow ODE (PF-ODE) as mentioned by previous works[42], which is actually deterministic given the sampled Gaussian noise is fixed. Here we designed an experiment which explores the performance change when using different i.i.d. Gaussian noise as input. Specifically, using the Euler solver, we evaluated the semantic occupancy forecasting model with 5 random seeds and recorded the average and standard deviation for IoU and mIoU across the three-second forecasting horizon (on the validation data).

We present the results in Table 23, and note that the standard deviation of the performance metrics for all three models is less than 1% of the mean across the entire forecasting horizon. Therefore, we conclude that our CFM model has low performance sensitivity to randomization in the sampling process and the uncertainty around the performance metrics evaluated is minimal.

Table 23: Different CFM models and their average IoU/mIoU (±std) over a three-second horizon, tested with 5 different random seeds.

| Model trained on | IoU↑ | | | | mIoU↑ | | | |
|---|---|---|---|---|---|---|---|---|
| | 1s | 2s | 3s | Avg | 1s | 2s | 3s | Avg |
| $O_v$ | 26.67±0.02 | 21.56±0.01 | 18.26±0.02 | 22.26±0.02 | – | – | – | – |
| $o_d$ | 39.82±0.13 | 29.34±0.04 | 23.73±0.07 | 30.93±0.01 | – | – | – | – |
| $O_s$ | 40.64±0.10 | 30.31±0.06 | 24.21±0.17 | 31.72±0.04 | 33.57±0.28 | 21.15±0.05 | 14.99±0.46 | 23.25±0.06 |

## G    Limitation and future work

**Physical consistency.** Although our method achieves state-of-the-art performance across different time horizons, the issue of continuous consistency in generated videos remains unresolved. For foreground objects, our current solution only achieves smooth output by performing timing processing within the network structure, but in the generated scenes, foreground objects are still prone to discontinuities. As shown in Table 20, although our work has greatly improved the accuracy of the forecasted foreground objects, there is still no theoretical or design guarantee that the foreground object will be continuous in the output (i.e., it will not suddenly appear or disappear in a few frames).

**Multi-modality forecasting.** For a safe end-to-end autonomous driving system, it is still very difficult for the current occupancy world model to not only decode the occupancy map after obtaining the future representation, but also forecast the future multi-modal trajectories of surrounding agents. As we shown in Table 23, the randomness in flow matching or diffusion (originates from Gaussian noise), under a given condition, the future forecast outputs are in fact quite fixed. This may be solved via more detailed control for foreground and background generation. We hope that this work will provide an efficient implementation framework for solving this issue in the future and accelerate the progress of subsequent work.

## H    Visualization

### H.1    Samples from the pretrained model

Here we visualize the forecasting results for two of the foundational models that are pretrained. Specifically, visualization for sparse occupancy forecasting is shown in figure 10 and visualization for dense occupancy forecasting is shown in figure 11. Please check more visualizations on our project page.

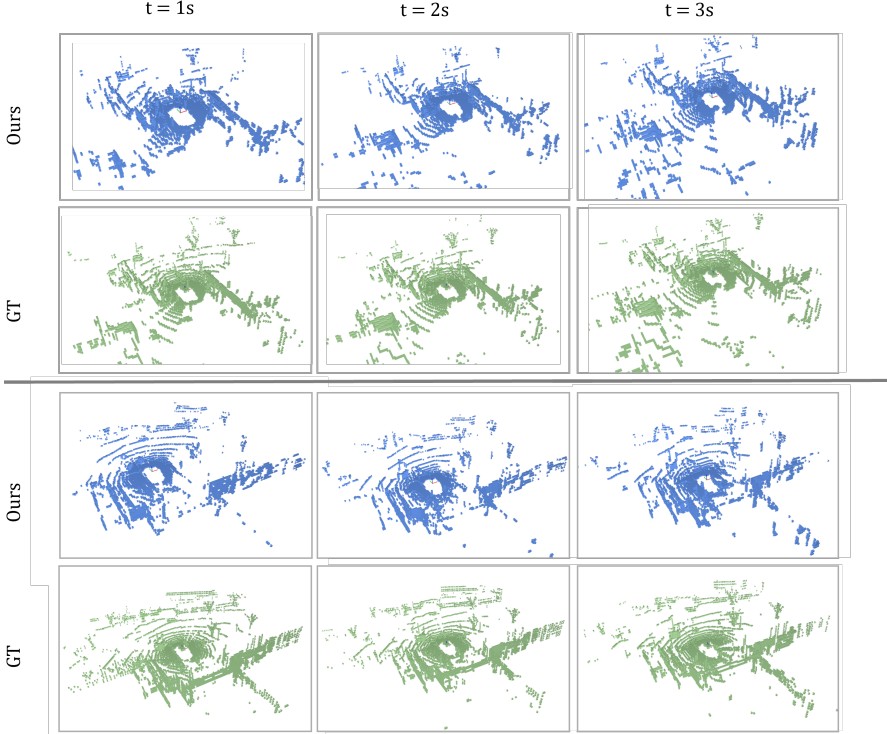

Figure 10: Visualization for the sparse occupancy($O_v$) forecasting over the three-second forecasting horizon.

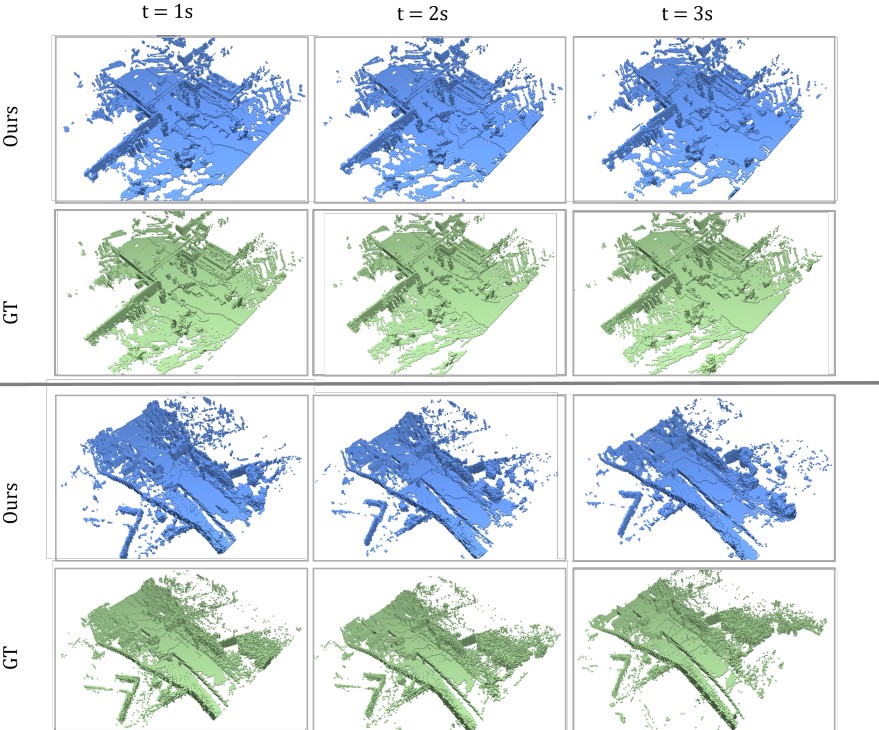

Figure 11: Visualization for the dense occupancy($o_d$) forecasting over the three-second forecasting horizon.

## H.2 Samples from the semantic occupancy forecasting model

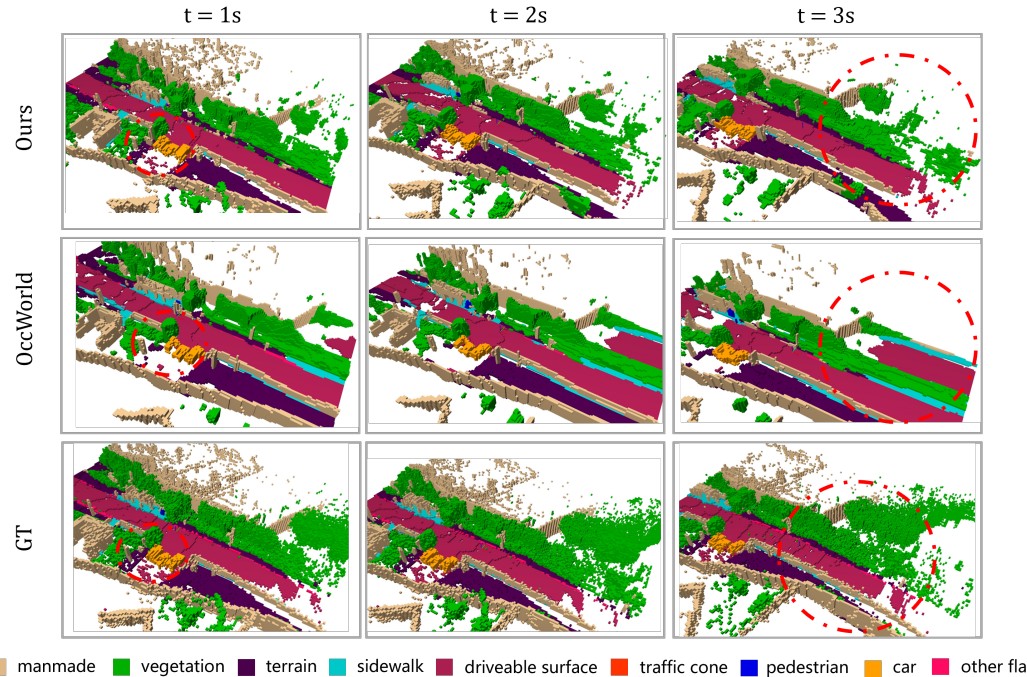

| manmade | vegetation | terrain | sidewalk | driveable surface | traffic cone | pedestrian | car | other flat |

Figure 12: Visualization for the semantic occupancy($O_s$) forecasting over the three-second forecasting horizon. Compared to the previous method, our method retains more details in this example (as shown in the red circle)

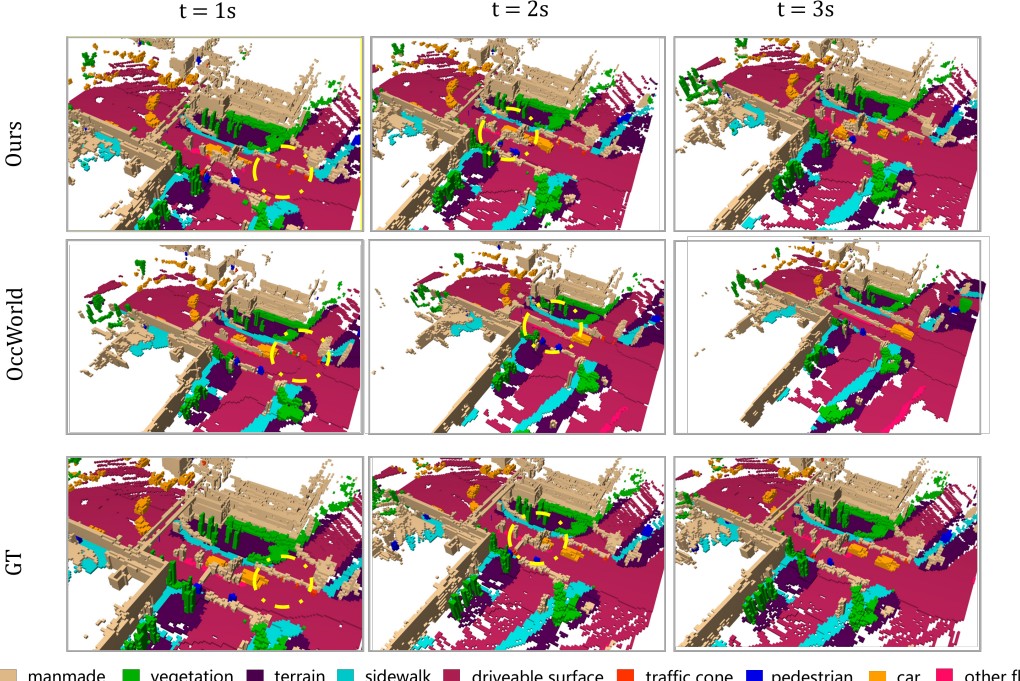

| manmade | vegetation | terrain | sidewalk | driveable surface | traffic cone | pedestrian | car | other flat |

Figure 13: Visualization for the semantic occupancy($O_s$) forecasting over the three-second forecasting horizon for an intersection. Compared to the previous method, as shown by the yellow circle, our methods better forecasts foreground (close-ranged) objects such as cars and barriers.

