# OpenReview forum: "Towards foundational LiDAR world models with efficient latent flow matching"
_NeurIPS.cc/2025/Conference — NeurIPS 2025 poster_

### Official Review · Reviewer_Tnf1 · 2025-06-29

**Clarity:** 2
**Significance:** 3
**Originality:** 3
**Rating:** 4
**Confidence:** 3

**Summary:**

This paper proposes a foundational LiDAR world model that can be  fine-tuned for various downstream tasks such as semantic occupancy forecasting and adapting to different LiDAR sensor configurations. The authors introduce a Swin Transformer-based VAE architecture to achieve state-of-the-art reconstruction accuracy with higher compression ratios and reduced computational costs. The model demonstrates transferability and reduced reliance on labeled data, achieving good performance with only a small fraction of the training data required by previous methods.

**Questions:**

See weakness.

**Ethical Concerns:**

["NO or VERY MINOR ethics concerns only"]

**Limitations:**

yes

**Paper Formatting Concerns:**

No formatting concern.

**Quality:**

3

**Strengths And Weaknesses:**

Strengths:

1.	The paper is well-written, with a clear methodology and extensive experimental results that validate its effectiveness.

2.	The motivation behind the module design is clearly articulated.

3.	The experiments are comprehensive. The authors demonstrate the effectiveness of their design across three challenging scenarios, achieving strong performance.

Weaknesses:

1.	The design primarily aims to address issues such as redundant model parameters and prolonged training time. However, it remains unclear why and how the proposed Swin Transformer-based VAE architecture effectively tackles these problems.

2.	Although the paper claims to be the first to introduce a foundational LiDAR world model, its future forecasting component appears to be quite similar to some existing self-supervised point cloud forecasting methods [1][2].

3.	Some descriptions and illustrations are not clear, and in some cases may be confusing. For instance, z_s^{t_1:t_2} is defined as the prediction target of the model in line 188 page 5, yet it is also shown as an input in Figure 3, which creates ambiguity.

[1] Yang, Z., Chen, L., Sun, Y., & Li, H. (2024). Visual point cloud forecasting enables scalable autonomous driving. In Proceedings of the IEEE/CVF Conference on Computer Vision and Pattern Recognition, 14673–14684.
[2] Khurana, T., Hu, P., Held, D., & Ramanan, D. (2023). Point cloud forecasting as a proxy for 4D occupancy forecasting. In Proceedings of the IEEE/CVF Conference on Computer Vision and Pattern Recognition, 1116–1124.

---

> ### Author Rebuttal · Authors · 2025-07-31
>
> We sincerely thank the reviewer for their thoughtful and detailed feedback, which helped us identify and address several important missing elements in our submission.
>
> > Re Q1:  "The design primarily aims to address issues such as redundant model parameters and prolonged training time. However, it remains unclear why and how the proposed Swin Transformer-based VAE architecture effectively tackles these problems."
>
> We are sorry for the misunderstandings that may have arisen from the narrative in our paper. Here we would like to recount the frame to help you better understand our thinking.
>
> In order to explore the adaption of the LiDAR world model on different subtasks and different data volumes, a challenge is prolonged training time: normally over a thousand GPU hours for SOTA model training. It is important to note that these training hours focus primarily on the training of the transition model(Figure 3), assuming a 2-stage training that follows the traditional Latent diffusion model.
>
> Excessive training time for transition models can be decomposed into 1. impropriety of the transition model training objective and 2. parameter redundancy of the transition model. For the former, we address it by replacing the original objective with a new CFM-based one via a redesigned DiT-based network(Figure 3). The related ablation study is shown in Table 4. For the latter, we found that it’s **actually correlated with the channel of latents**(128 in OccWorld and 64 in DOME), this motivates us to have a better data compressor for a higher compression ratio as we present in Figure 2. Therefore, the data compressor in fact **only solves the previous problem of low latent compression rates**. Its own training won't take less time than the data compressor in previous work, but **the low-dim latent it generates will help us reduce the parameters in the transition model**.
>
> Further, for the question “Why does your VAE work better in terms of information preserving?” Apart from the advantages of the structure itself (Swin Transformer v.s. All conv data compressors before), we attribute this to height embedding, removal of all 3D structures and further compression of the dimension at the bottleneck.
>
> Specifically, we first attribute this to the removal of all temporal processing modules: experimentally, our aggregation for spatial dimensional information at the transition model is sufficient, and adding 3D conv in decoder like DOME instead affects the reconstruction rate, as shown in the following table.
>
> | ×192 compression ratio          | IoU  | mIoU |
> |--------------------------------|-----:|-----:|
> | With 3-D conv + 3-D attention   | 82.33 | 88.75 |
> | Without (2-D only)              | 85.12 | 92.81 |
>
> With these 2 different kinds of latents, we retrain the model for comparsion in forecasting performance:
>
> | Variant | 1 s&nbsp;IoU | 1 s&nbsp;mIoU | 2 s&nbsp;IoU | 2 s&nbsp;mIoU | 3 s&nbsp;IoU | 3 s&nbsp;mIoU | **Avg IoU** | **Avg mIoU** |
> |---------|------------:|--------------:|------------:|--------------:|------------:|--------------:|------------:|-------------:|
> | With ~  | 39.72 | 31.68 | 28.77 | 20.00 | 21.14 | 13.91 | **29.87** | **21.86** |
> | W.o. ~  | 40.53 | 33.17 | 30.37 | 21.09 | 24.44 | 15.64 | **31.78** | **23.33** |
>
> At the same time, we note the importance of the Layer-norm block (LN+Conv+ Swish Activation) before sampling and height embedding, as shown in the Table below. The former contributes to scale stabilisation of latent whereas the latter has been overlooked from time to time by previous work (they borrowed directly from the data compressor for stable diffusion, whereas the height dimension does not exist in video data).
>
> | Component              | Variant  | IoU   | mIoU |
> |------------------------|----------|------:|-----:|
> | LN block               | With     | 83.62 | 89.80 |
> |                        | W.o.     | 85.12 | 92.81 |
> | Height embedding       | With     | 81.92 | 88.63 |
> |                        | W.o.     | 85.12 | 92.81 |
>
> These ablation study in VAE work mechanism will be updated in the appendix. We hope that these tricks will help later work on designing networks more efficiently, and these answers will answer your questions.
>
> > Re Q2: Although the paper claims to be the first to introduce a foundational LiDAR world model, its future forecasting component appears to be quite similar to some existing self-supervised point cloud forecasting methods [1][2].
>
> We would like to make two points here to show the difference between our method and ViDAR, and why we did not compare the results of point cloud forecasts in the paper.
>
> 1. We would like to emphasise that by utilising 4D prior knowledge from large-scale pre-training, we can reduce the amount of data used in the downstream world model task(dynamic learning), especially for non-semantic to semantic transferability. By contrast, the contribution of ViDAR here is to improve the performance of perceptual(3D)/end-to-end(4D) autonomous driving that requires labels, with the backbone pretrained via self-supervised visual point cloud forecasting. The 4D tasks benefit from ViDAR, like the future occupancy forecasting (non-semantic, Table 8 in ViDAR) results they reported, is based on fine-tuning ViDAR pre-trained UniAD, while the fine-tuning of UniAD needs labels. Therefore it is inappropriate to directly compare the IOUs of the point cloud forecast by our method (Table 2) with Table 8 in ViDAR, since our training does not require future labels. We admit that theoretically ViDAR can also forecast semantic occupancy, but this is not implemented in the article or in the publicly available codes.
>
> 2. Rendering the point cloud in real-time and calculating the gradient in forward propagation is very memory-intensive and time-consuming(in Occ4d[1] implementation, it needs over 20G memory for the PC rendering in a 800x800 voxel space), and given our real-world situation we use voxel- rather than point-based forecasting. This reduction in granularity allows our model to be faster than the previous baseline model like OccWorld(Deterministic Transformer based) or DOME (DDPM based).
>
> While it is not practical to directly compare results after pre-training, we note that we can indeed compare results from pre-training. Again, we thank the reviewers for pointing out this point, which was previously overlooked. For simplicity, we voxelise all of the ViDAR and Occ4d forecasts to calculate the IOUs and compare them with our approach. Since ViDAR only releases the weight pretrained on ⅛ nusc for 3s future forecasting(the line we marked with star in following tables), but not the one they mentioned in the paper, we use it as an instance here. From the chamfer distance perspective, the one pretrained on ⅛ of nusc is close to the one on full dataset.
>
> (ViDAR results)
> |    | 1s     | 2s     | 3s     |
> |---------------|--------|--------|--------|
> | L1 error*       | 2.541  | 2.740  | 2.990  |
> | Abs rel error*  | 0.188  | 0.209  | 0.244  |
> | Chamfer*       | 1.251  | 1.489  | 1.790  |
> | Chamfer in paper, full dataset      | 1.120  | 1.380  | 1.730  |
> | IOU*           | 13.11  | 12.48  | 11.78   |
>
> (Occ4d results)
> | | 1s     | 2s     | 3s     |
> |---------------|--------|--------|--------|
> | L1 error      | 1.44 | 1.80 | 2.16 |
> | Abs rel error | 0.10 | 0.14 | 0.18 |
> | Chamfer       | 1.11   | 1.45   | 1.96   |
> | IOU           | 26.96  | 19.51  | 16.81  |
>
> Compared to these methods, our performance remains ahead of the curve, especially at 2s and 3s forecast horizon, as shown in the following table.
> |     | 1s     | 2s     | 3s     |
> |-----------------|--------|--------|-------|
> | ViDAR* (Img2LiDAR)     | 13.11  | 12.48  | 11.78  |
> | Occ4d (LiDAR2LiDAR)     | 26.96  | 19.51  | 16.81  |
> | Ours            | **26.98** | **21.56** | **18.26** |
>
> Our method exceeds the Occ4d and also ViDAR in the pretraining stage, this can also be another aspect of the effective learning of the dynamic prior during our pre-training (without semantics). Another noteworthy point is that for both ViDAR and Occ4d, the prediction of the point cloud requires every ray angle from the future point cloud (which can be obtained from the vehicle's future pose and sensor's internal parameters), and the network is only responsible for forecasting the depth, whereas our approach here does not require any information from the future.
>
>
> > Re Q3: "Some descriptions and illustrations are not clear, and in some cases may be confusing. For instance, $z_s^{t_1:t_2}$ is defined as the prediction target of the model in line 188 page 5, yet it is also shown as an input in Figure 3, which creates ambiguity."
>
> Thank you for pointing out the ambiguity. We use time point $t_1$ as the demarcation point when the prediction started, illustrated on line 188. The historical latent $z_s^{t_0:t_1}$ stand for encoded past frames and is fed into the network unchanged, as the condition. The training of flow matching need us to construct a noised future ground truth, act as a training sample along the probability path between prior distribution to the data distribution. Therefore, we have to use $z_s^{t1:t2}$ in Figure 3. However, in sampling process(test stage), the $x_t$ here will be pure Gaussian noise, so the future infos won't be leaked. We understand it seems ambiguious that we use the "objective" in training stage, we will update a more detailed description here.

---

### Official Review · Reviewer_br6R · 2025-06-30

**Clarity:** 3
**Significance:** 4
**Originality:** 4
**Rating:** 5
**Confidence:** 3

**Summary:**

The paper introduces the first general-purpose LiDAR world model with key contributions on 1) A Swin-Transformer VAE compresses voxelised LiDAR grids by 192 × while preserving ≥ 93 % mIoU. 2) A Conditional Flow-Matching UNet predicts future latent frames in real time (~22 FPS), needing far fewer FLOPs than diffusion baselines. 3) Pre-training on nuScenes lets the model fine-tune with only 5 % labelled data (for sparse-to-dense completion, outdoor-to-indoor forecasting, and semantic occupancy), beating task-specific SOTA. 4) A cosine-similarity regulariser keeps latent codes aligned when the encoder is replaced, safeguarding transfer.

**Questions:**

Question: Why is outdoor-to-indoor the right showcase for “cross-domain” generalisation when indoor scenes are typically easier and better labelled?
Suggestion: Please provide quantitative evidence—e.g., annotation cost comparison, data-scarcity statistics, or failure cases—showing that indoor fine-tuning really benefits from the outdoor pre-training, and clarify why the reverse direction is less meaningful.
Impact on score: A convincing justification (or an alternate, harder transfer task) would remove my main conceptual doubt and could raise the score to 6/6.

**Ethical Concerns:**

["NO or VERY MINOR ethics concerns only"]

**Final Justification:**

After carefully reviewing the other reviewers’ discussions, I will maintain my original rating.

**Limitations:**

yes

**Paper Formatting Concerns:**

no Paper Formatting Concerns

**Quality:**

3

**Strengths And Weaknesses:**

Strengths

1. Originality: The authors are the first to package foundation-model style pre-training, compression, and cross-task transfer into a single LiDAR-only world model. No previous work nails the whole pipeline end-to-end.
2. Quality: Experiments are thorough: 192× compression still keeps ≥ 93 % mIoU, ablations cleanly isolate the VAE, flow-matching, and latent-alignment tricks, and real-time numbers (≈ 22 FPS) are reported on the same hardware as baselines.
3. Significance: Needing only 5 % labelled data to beat task-specific SOTA on three diverse tasks is a big win for anyone strapped for annotations.
4. Clarity: Figures are self-explanatory, training details live in the appendix, and the writing flows—you rarely re-read a sentence to “get it.”



Weaknesses

I am not persuaded by the motivation of the proposed downstream task "outdoor to indoor generalisation", since intuitively, the outdoor is a much more challenging scenario and suffers more from data annotation scarcity compared to indoor; it would be helpful if the author could provide more empirical evidence to strengthen the motivation. Additionally, the introduction appears to be overly long, and some typos, such as those on line 7, should be carefully checked.

---

> ### Author Rebuttal · Authors · 2025-07-31
>
> We would first like to thank the reviewer for their feedback. The insights and questions you kindly offered regarding the nature and motivation for the outdoor-to-indoor navigation task are helpful to improve the quality of our paper. Below we give a detailed explanation of why we choose outdoor-to-indoor navigation as one of our cross-domain experiments, why it is practically relevant, and why the reverse direction is less meaningful.
>
> > Re: “Why is outdoor-to-indoor the right showcase for “cross-domain” generalisation when indoor scenes are typically easier and better labelled?” & “showing that indoor fine-tuning really benefits from the outdoor pre-training.”
>
> We would like to clarify why we frame outdoor-to-indoor as the transfer direction. Althought both directions are interesting, outdoor-to-indoor is the more practical direction due to the diversity and scale of existing LiDAR data. Outdoor LiDAR datasets are a. Plentiful, freely available, and well-calibrated: many are already aligned with GPS/IMU and camera sensors. b. Rich in dynamic behavior: diverse scenes, agents, and motion profiles.
>
> Outdoor environments exhibit a wider range of dynamic behaviors that are essential for training a transferable world model in many aspects: High-momentum ego motion: a vehicle may travel tens of meters and rotate several degrees between 0.5s sweeps, producing strong range-rate cues and frequent self-occlusion. Articulated actors: buses articulate, cyclists lean, trailers pivot, capturing realistic temporal dynamics. Long-range occlusions: objects can disappear behind others and reappear several frames later. Mixed mobility: static poles, parked cars, and fully dynamic agents co-exist, and must be disambiguated from geometry alone. Training on such diverse outdoor motion helps establish strong dynamics priors that generalize across domains.
>
> Our goal is to test whether such motion priors learned from highly dynamic outdoor environments can transfer to indoor scenes, which feature:
> - Short-range occlusions (2–4 m instead of 40 m),
> - Dense LiDAR returns and cluttered geometry
> - Slow, fine-grained human and robot motion,
> - Rapid local changes in confined space.
>
> In summary, we believe the dynamics prior in outdoor environments is more general and could be more suitable in a more complex scenarios.
>
> > Re: “ clarify why the reverse direction is less meaningful.”
>
> We believe the reverse direction would be interesting to test, and not necessarily less meaningful; we will make this clear in the paper. However, it is simply less practical to attempt this transfer direction for several reasons. First, indoor datasets are scarcer. Second, motion information embedded in indoor environments is, in some sense, not as diverse as that in outdoor scenes. Motion typically happens in small ranges and the variation in objects is limited.

---

> > ### Comment · Reviewer_br6R · 2025-08-06
> >
> > Thank you for the thorough explanation—it provides valuable insights. After carefully reviewing the other reviewers’ discussions, I will maintain my original rating.

---

### Official Review · Reviewer_qmHa · 2025-06-30

**Clarity:** 3
**Significance:** 3
**Originality:** 3
**Rating:** 4
**Confidence:** 3

**Summary:**

This paper proposes a foundational LiDAR world model trained via latent flow matching on large-scale LiDAR sequences. It combines a Swin-VAE for efficient BEV occupancy compression and a conditional flow matching decoder for future occupancy prediction. The model is pretrained in a self-supervised manner and evaluated on multiple downstream tasks, showing strong generalization under low-data regimes and significant efficiency gains.

**Questions:**

1. Why is Visual Point Cloud Forecasting (CVPR 2024) not discussed or compared, given that it also uses point cloud forecasting as a self-supervised pretraining objective?
2. Figures 2 and 3 illustrate key components of your method but are never referenced in the main text. Can you clarify why they were omitted from the narrative, and how readers are expected to connect them with Sections 3.2 and 3.3?
3. As a LiDAR world model, does your method support explicit reconstruction of future point clouds? If so, how do you evaluate the geometric quality of these outputs beyond voxel mIoU—for example, in terms of fidelity, completeness, or temporal consistency?

**Ethical Concerns:**

["NO or VERY MINOR ethics concerns only"]

**Final Justification:**

The rebuttal has addressed my main concerns regarding the comparison with ViDAR, clarification of the methodology, and evaluation metrics. The additional results and explanations resolve my doubts, and I therefore maintain my positive recommendation.

**Quality:**

3

**Strengths And Weaknesses:**

**Strengths:**
1. The paper introduces a pretraining architecture that combines a Swin-VAE for BEV compression with a latent flow matching module for temporal modeling. This framework enables scalable self-supervised learning over large-scale LiDAR sequences and supports transfer across multiple forecasting tasks.
2. The pretrained model significantly reduces reliance on human labels. For example, on semantic occupancy forecasting, it outperforms supervised baselines using only 5% of labeled data, indicating strong inductive priors learned from geometric dynamics.
3. The Swin-VAE achieves up to 6× higher compression than prior work while maintaining high reconstruction fidelity. Combined with latent-space forecasting, the method runs up to 28.2× faster and requires substantially fewer FLOPs than baselines.

**Weaknesses:**
1. The paper does not compare to recent works like Visual Point Cloud Forecasting (CVPR'24), which also use point cloud forecasting as a pretraining objective. Without conceptual or empirical comparison, the novelty claim of being the “first foundational LiDAR world model” lacks proper grounding.
2. Figures 2 and 3 are not referenced in the main text, making it harder to follow how temporal modeling is implemented in practice.
3. Although the model aims to predict future LiDAR scenes, it does not evaluate the geometric or structural quality of the predicted outputs. There is no analysis of consistency, plausibility, or failure modes, leaving the realism and utility of the predictions unassessed.

---

> ### Author Rebuttal · Authors · 2025-07-31
>
> Thank you for pointing out our shortcomings in comparison and experiment design! We provide a more detailed comparison and analysis here.
>
> > Re Q1:  “Why is Visual Point Cloud Forecasting (CVPR 2024) not discussed or compared, given that it also uses point cloud forecasting as a self-supervised pretraining objective?”
>
> We’re sorry if our statement caused some misunderstanding. We would like to make two points here to show the difference between our method and ViDAR, and why we did not compare the results of point cloud forecasts in the paper.
>
> 1. We would like to emphasise that by utilising 4D prior knowledge from large-scale pre-training, we can reduce the amount of data used in the downstream world model task(dynamic learning), especially for non-semantic to semantic transferability. By contrast, the contribution of ViDAR here is to improve the performance of perceptual(3D)/end-to-end(4D) autonomous driving that requires labels, with the backbone pretrained via self-supervised visual point cloud forecasting. The 4D tasks benefit from ViDAR, like the future occupancy forecasting (non-semantic, Table 8 in ViDAR) results they reported, is based on fine-tuning ViDAR pre-trained UniAD, while the fine-tuning of UniAD needs labels. Therefore it is inappropriate to directly compare the IOUs of the point cloud forecast by our method (Table 2) with Table 8 in ViDAR, since our training does not require future labels. We admit that theoretically ViDAR can also forecast semantic occupancy, but this is not implemented in the article or in the publicly available codes.
>
> 2. Rendering the point cloud in real-time and calculating the gradient in forward propagation is very memory-intensive and time-consuming(in Occ4d[1] implementation, it needs over 20G memory for the PC rendering in a 800x800 voxel space), and given our real-world situation we use voxel- rather than point-based forecasting. This reduction in granularity allows our model to be faster than the previous baseline model like OccWorld(Deterministic Transformer based) or DOME (DDPM based).
>
> While it is not practical to directly compare results after pre-training, we note that we can indeed compare results from pre-training. Again, we thank the reviewers for pointing out this point, which was previously overlooked. For simplicity, we voxelise all of the ViDAR and Occ4d forecasts to calculate the IOUs and compare them with our approach. Since ViDAR only releases the weight pretrained on ⅛ nusc for 3s future forecasting(the line we marked with star in following tables), but not the one they mentioned in the paper, we use it as an instance here. From the chamfer distance perspective, the one pretrained on ⅛ of nusc is close to the one on full dataset.
> (ViDAR results)
> |    | 1s     | 2s     | 3s     |
> |---------------|--------|--------|-------|
> | L1 error*       | 2.541  | 2.740  | 2.990  |
> | Abs rel error*  | 0.188  | 0.209  | 0.244  |
> | Chamfer*       | 1.251  | 1.489  | 1.790  |
> | Chamfer in paper, full dataset      | 1.120  | 1.380  | 1.730  |
> | IOU*           | 13.11  | 12.48  | 11.78  |
>
> (Occ4d results)
> | | 1s     | 2s     | 3s     |
> |---------------|--------|--------|-------|
> | L1 error      | 1.44 | 1.80 | 2.16 |
> | Abs rel error | 0.10 | 0.14 | 0.18 |
> | Chamfer       | 1.11   | 1.45   | 1.96   |
> | IOU           | 26.96  | 19.51  | 16.81  |
>
> Compared to these methods, our performance remains ahead of the curve, especially at 2s and 3s forecast horizon, as shown in the following table.
> |     | 1s     | 2s     | 3s     |
> |-----------------|--------|--------|-------|
> | ViDAR* (Img2LiDAR)     | 13.11  | 12.48  | 11.78  |
> | Occ4d (LiDAR2LiDAR)     | 26.96  | 19.51  | 16.81  |
> | Ours            | **26.98** | **21.56** | **18.26** |
>
> Our method exceeds the Occ4d and also ViDAR in the pretraining stage, this can also be another aspect of the effective learning of the dynamic prior during our pre-training (without semantics). Another noteworthy point is that for both ViDAR and Occ4d, the prediction of the point cloud requires every ray angle from the future point cloud (which can be obtained from the vehicle's future pose and sensor's internal parameters), and the network is only responsible for forecasting the depth, whereas our approach here does not require any information from the future.
>
> > Re: “Figures 2 and 3 illustrate key components of your method but are never referenced in the main text. Can you clarify why they were omitted from the narrative, and how readers are expected to connect them with Sections 3.2 and 3.3?”; "... harder to follow how temporal modeling is implemented in practice."
>
> We thank the reviewer for pointing out this inconsistency. We have added references and more explanations of them in the main text. The two figures were indeed intended to complement the technical exposition in Section 3.1 (Data compression) and 3.2 (Conditional Flow Matching, CFM). We clarify the connection below.
>
> Figure 2 is an overview of the Swin-VAE encoder/decoder architecture which is described in Section 3.1 and Figure 3 depicts specific architectural design for our conditional velocity field predictor that comes immediately after the encoder. We expect the reader to connect these two figures with the description of our methodology. Basically we need to first train a VAE to compress the data, then use the compressed latent to forecast future latent (via CFM), finally decode the future observation. In the training of CFM, the VAE is frozen.
>
> For Section 3.3 (Representation alignment), we explained how we align the latent space of VAE (described in Figure 2 and Section 3.1) when performing cross-domain finetuning (e.g. dense occupancy → semantic occupancy) with more details on related visualization and experimental results in Appendix D. We will add references for clarity. Specifically, we found that when the non-semantic pre-trained CFM weights are directly used to fine-tune the latent with semantic information, the results are not optimal. After comparing CKA and CKNNA we found that the hidden space of both (latent w/w.o. semantic information) is not aligned. To solve this problem, we introduced a cosine similarity term (Eq. 2) to encourage the alignment of latent w/w.o. semantic information in the space when fine-turning the data compressor for semantic occupancy (as shown in Figure 6). The other reason for this is that in practice, because of the difference in embedding, we cannot simply fine-tune the VAE for non-semantic data with semantic data. This alignment proves to be effective, and we can obtain further improvements of 2 to 6 per cent after alignment (as shown in Table 8).
>
> Further, when the pre-training data and the fine-tuning data were not ‘paired’, we simply fine-tuned the representation of the data compressor on the fine-tuning data instead of directly using the weights obtained on the pre-training data. Experiments have also demonstrated the effectiveness of this fine-tuning(in Table 6 and 7).
>
> Therefore, the overall experimental flow is: 1. Use the pre-trained data to train the VAE as a data compressor. 2. Use this VAE to obtain the latents of the pre-trained data. 3. Use these latents to pre-train the CFM. 4. Fine-tune this VAE on the downstream data (either directly or by using the cos term to align). 5. Fine-tune the CFM on the downstream.
>
> > Re: As a LiDAR world model, does your method support explicit reconstruction of future point clouds? If so, how do you evaluate the geometric quality of these outputs beyond voxel mIoU—for example, in terms of fidelity, completeness, or temporal consistency?
>
> Our approach supports explicit point cloud rendering, but given the computational cost and practical usefulness we don't do it, so that's why our last rendering step in the decoder section of Figure 2 is dashed.
>
> Re: evaluating geometric quality: for fidelity, completeness, and temporal consistency, following DynamicCity [2] and LiDM [3], we use latents from Minkovski-Unet for fidelity measurement (FID/KID) and trained a 4D semantic occupancy network based on our VAE for temporal consistency(FVD) measurement. Specifically, for FID/KID:
>
> |        | 1s FID | 1s KID | 2s FID | 2s KID | 3s FID | 3s KID | AVG FID | AVG KID |
> |--------------------|--------|--------|--------|--------|--------|--------|---------|---------|
> | OccWorld           | 9.54   | 11.7   | 8.56   | 10.46  | 7.78   | 9.80   | 8.62    | 10.60   |
> | DOME               | 4.37   | 4.39   | 5.09   | 5.44   | 5.63   | 6.53   | 5.03    | 5.42    |
> | Ours (Hist. traj)  | 1.67   | 1.49   | 2.81   | 2.85   | 4.02   | 4.74   | 2.83    | 3.02    |
> | Ours (Fut. traj)   | 1.61   | 1.47   | 2.90   | 3.05   | 3.91   | 4.55   | **2.80**    | **3.02**    |
>
> For FVD:
> Based on our VAE, we include 3D conv and time attention in both the encoder and decoder to ensure that the network can capture the temporal relation. The reconstruction rate of the network used as occupancy video feature extractor is 81.39 and 87.75 in terms of IOU and mIOU. The FVDs are shown below (3s, 6 frames). To give context for temporal consistency, we include an approach that randomly shuffles the ground truth future ordering and compare the FVD with the correctly ordered GT. This tells us the FVD when the predictions are individually correct but temporally inconsistent.
>
> |         | FVD (×10⁻³) |
> |-------------------|---------|
> | OccWorld          | 18.68       |
> | DOME              | 9.79        |
> | Ours (hist. traj.)| 7.80        |
> | Ours (fut. traj.)| **7.68**        |
> | Reorder GT        | 12.07        |
>
> Finally we would like to thank the reviewers again for help and hope that our answers will solve your problems and provide sufficient reason to raise the score.
>
> [1] Point Cloud Forecasting as a Proxy for 4D Occupancy Forecasting, CVPR2023
>
> [2] DynamicCity: Large-Scale 4D Occupancy Generation from Dynamic Scenes, ICLR2025
>
> [3] LiDAR Diffusion Models, CVPR2024

---

> > ### Comment · Reviewer_qmHa · 2025-08-06
> >
> > Thank you for the clarifications and the additional experiments. I don’t have further questions. For the final version, it would be great if you could address the points I raised earlier and integrate the new results from the tables, which I believe would make the paper clearer and more convincing.

---

### Official Review · Reviewer_N8qa · 2025-07-03

**Clarity:** 3
**Significance:** 3
**Originality:** 2
**Rating:** 4
**Confidence:** 3

**Summary:**

The paper explores building foundational LiDAR world models, transferability to downstream forecasting tasks and efficient data compression via CFM.

**Questions:**

1. What are the challenges of buiding a foundational LiDAR world model and how do you address them? The reasoning is missing in the introduction. Alternatively, can augmenting a powerful, existing image-based foundational model (like GAIA-2 or Cosmos) with a LiDAR-based perception module also achieve your goal? Could you provide evidence or a stronger argument for why a monolithic LiDAR world model is fundamentally superior to a hybrid system that combines existing visual world model with a LiDAR perception module for geometric accuracy? What critical capabilities are lost in such a modular approach?

2. Is it proper to state the learned representation as "universal 3D motion prior"? What specific properties of your learned representation suggest it is universal rather than just a highly effective prior for ground vehicle navigation?

3. Could you provide an analysis/discussion of forecasting performance as a function of the VAE's compression ratio? At what point does the information loss from compression begin to significantly degrade the model's ability to predict complex or subtle dynamic events? This is crucial for understanding the model's limitations in safety-critical scenarios.

**Ethical Concerns:**

["NO or VERY MINOR ethics concerns only"]

**Final Justification:**

I am satisfied with the rebuttal, so I choose to maintain the positive rating.

**Limitations:**

1. Performance in Non-Ideal Conditions: The model is trained and evaluated on well-curated datasets. Its robustness to real-world sensor noise, artifacts, and adverse weather conditions (e.g., rain, fog, snow), which significantly degrade LiDAR quality, is unknown and unaddressed.

2. Safety considerations: For a model intended for safety-critical applications like autonomous driving, an analysis of its failure modes is essential. The paper lacks any discussion of when the model's forecasts are likely to be inaccurate or unreliable. Understanding these failure cases is critical for safe deployment.

3. Potential for Malicious Use: A powerful generative model capable of creating realistic 4D representations of real-world scenes could potentially be used to mislead or attack autonomous driving systems.

**Quality:**

3

**Strengths And Weaknesses:**

**Strengths**:
- the "foundational" model for LiDAR-based world modeling could be of interest to relevant community

**Weaknesses**:
- The "Foundational" Claim seems Overstated. The paper uses transfer learning success as evidence of a "foundational" model that has learned a "universal 3D motion prior." However, the pre-training (outdoor driving) and all fine-tuning tasks (different LiDAR beams, indoor navigation, semantic driving scenes) fall within the domain of ground-level navigation. The current results of a large model pre-trained on a relevant, large dataset performs well on similar downstream tasks, appears not necessarily evidence of a foundational model.
- Trade-offs in compression efficiency and forecasting. The paper's focus on high-ratio compression (up to 768x) is a key part of its efficiency claim. However, there is a lack of analysis on what information is lost during this compression and how that impacts the model's ability to learn nuanced dynamics. High compression might filter out subtle cues (e.g., a slight change in a pedestrian's posture) that are critical for predicting complex, long-term events. The model may be efficient at forecasting general motion but could be brittle in safety-critical edge cases, a trade-off that is never explored.

---

> ### Author Rebuttal · Authors · 2025-07-31
>
> Thank you for your thoughtful review and helpful feedback, we appreciate it. Please see our per-question responses below.
>
> > Re: Q1.1: What are the challenges of building a foundational LiDAR world model? The reasoning is missing in the introduction.
>
> Thank you for your comment, we will include a discussion of the challenges and our approach to them in the introduction near lines 36. Here is specific respond:
>
> The main challenges are the 1. **availability of the data** and 2. the **costs of training**. We will include these contents in the final version to better demonstrate our motivation.
> 1. As you mentioned, Cosmos and GAIA-2 often require millions of GPU hours to train and the results of the training are built on sufficient data: Cosmos used 100 million RGB video clips with diverse content for pre-training; assuming 10s per clip, this equates to about 277k hours of pre-training data (see page 7 of the Cosmos technical report). However, this diversity and scale is not reproducible with existing public dataset of LiDAR: Argoverse2, as the largest open LiDAR dataset, only provides around 166 hours of data of exclusively autonomous driving scenes. Our approach involves two aspects to address this issue. First, we simplified the task itself from building a ‘foundational model’ that is pre-trained on an all-encompassing dataset to one which is trained on data that is easy to be obtained and used in a data-poor domain, which is one of the reasons why we use ‘Towards’ in the title. On the other hand, considering the lack of lidar data diversity, we collected an indoor lidar dataset ourselves (about 4 hour length) to explore the transferability on large gap domains.
> 2. For dynamic learning, most of the previous LiDAR world models require a few thousands epochs of training, which may take a few thousand GPU hours.  We designed an efficient approach that required fewer than 200 epochs (about 14 hours on 2x 5090) for the network to converge and achieve the SOTA performance on Voxel-based LiDAR forecasting, as we shown in Table 2.
>
> > Re: Q1.2 “Can augmenting a powerful, existing image-based foundational model (like GAIA-2 or Cosmos) with a LiDAR-based perception module also achieve your goal?”
>
> Thank you for the thoughtful question, this is definitely an interesting direction for future exploration and comparison to our proposed approach. Intuitively, augmenting Cosmos or GAIA with LiDAR could help the model as RGB images provide a lot of semantic information. However, there are some practical considerations: Cosmos and GAIA can generate high-fidelity samples but the model parameters are up to 7B/14B (ours is 30.3M) and the inference time per video sequence is in minute-scale, which makes real-time forecasting difficult.  We tested with the smallest version of cosmos (2B, video2world, bf16 precision) on the L40s, requires 32.6GB of video memory and about 4.5 minutes to produce a 5-second video, while our model achieves about 24FPS with only 2.35GB memory in the same environment. Obtaining future observation is only a precursor to generating control signals, and the latency at this stage is already high.
>
> Furthermore, fine-tuning a world model such as Cosmos using LiDAR could be  challenging. The essential reason is that we lack accurate LiDAR depth for self-supervision in arbitrary scenarios (although it may be possible to provide pseudo labelling through models like Depth Anything,).
>
> > Re Q1.3 and Q1.4: “Could you provide evidence or a stronger argument for why a monolithic LiDAR world model is fundamentally superior to a hybrid system” and “What critical capabilities are lost in such a modular approach?”
>
> While we agree it would be interesting to approach LiDAR forecasting with a hybrid world model, our paper does not claim that the LiDAR world model would be fundamentally superior to this hypothetical hybrid system. An interesting fact is that on some mature perception tasks (e.g. 3D detection/segmentation), SOTA's approach almost invariably uses a LiDAR/RGB fusion scheme. As mentioned before, one of the main problems with doing this in the world model is that not any of the scenarios are accurately depth-referenced, but we still believe this will be one of the possible directions for future development.
>
>
> > Re Q2: “Is it proper to state the learned representation as "universal 3D motion prior"? What specific properties of your learned representation suggest it is universal rather than just a highly effective prior for ground vehicle navigation?”
>
> One piece of evidence for this is that in the indoor data, although the platform of LiDAR sensor is still a ground vehicle, the scope of the scene and the movement of foreground objects is completely different from the outdoor scene. The movement of the crowd is more complex, and the environment changes from an open background to a closed scene. Our framework still has performance gain by finetuning the foundational model that is trained on unlabeled outdoor data.
>
> Nevertheless, we agree that the qualifier "for ground vehicles" is needed. We did not add this point to the explicit contributions statement originally, because it is a broader goal of the line of work, as opposed to an explicit contribution of this paper. We will make this clear in the paper.
>
> > Re Q3: “Could you provide an analysis/discussion of forecasting performance as a function of the VAE's compression ratio? At what point does the information loss from compression begin to significantly degrade the model's ability to predict complex or subtle dynamic events? This is crucial for understanding the model's limitations in safety-critical scenarios.”
>
> Here is the case of using VAEs with different compression ratios as data compressor for the model: we first analysed which part of the scene would be more likely to be lost in the compression process, as shown in the table below. Note that here we use bfp16 acceleration for FPS measurement, which will have about 0.5% to 1.4% negative impact on performance compared with the one reported in the paper.
>
> (Drop over 30% are bolded)
> | Category              | x32  | x192  | x384  | x768  | Drop&nbsp;(x32→x192) | Drop&nbsp;(x32→x384) | Drop&nbsp;(x32→x768) |
> |-----------------------|------:|------:|------:|------:|----------------:|----------------:|----------------:|
> | others                | 98.56 | 88.51 | 79.56 | 66.14 | 10.05 | 19.00 | **32.42** |
> | barrier               | 99.38 | 97.80 | 95.14 | 88.74 |  1.58 |  4.24 | 10.64 |
> | bicycle               | 98.61 | 97.45 | 96.03 | 90.77 |  1.16 |  2.58 |  7.84 |
> | bus                   | 99.54 | 93.63 | 87.75 | 79.57 |  5.91 | 11.79 | 19.97 |
> | car                   | 99.68 | 96.00 | 88.94 | 78.89 |  3.68 | 10.74 | 20.79 |
> | construction_vehicle  | 97.69 | 84.19 | 75.26 | 62.37 | 13.50 | 22.43 | **35.32** |
> | motorcycle            | 99.59 | 98.42 | 96.34 | 90.49 |  1.17 |  3.25 |  9.10 |
> | pedestrian            | 99.46 | 97.65 | 95.28 | 88.07 |  1.81 |  4.18 | 11.39 |
> | traffic_cone          | 99.07 | 97.68 | 95.33 | 87.73 |  1.39 |  3.74 | 11.34 |
> | trailer               | 98.84 | 89.12 | 81.89 | 69.97 |  9.72 | 16.95 | 28.87 |
> | truck                 | 99.21 | 92.71 | 85.67 | 76.63 |  6.50 | 13.54 | 22.58 |
> | driveable_surface     | 99.71 | 97.27 | 93.01 | 87.55 |  2.44 |  6.70 | 12.16 |
> | other_flat            | 99.59 | 98.56 | 96.61 | 90.74 |  1.03 |  2.98 |  8.85 |
> | sidewalk              | 99.21 | 94.40 | 86.94 | 77.29 |  4.81 | 12.27 | 21.92 |
> | terrain               | 98.83 | 93.86 | 86.61 | 76.03 |  4.97 | 12.22 | 22.80 |
> | manmade               | 97.87 | 83.66 | 72.52 | 62.53 | 14.21 | 25.35 | **35.34** |
> | vegetation            | 93.66 | 68.27 | 58.32 | 50.56 | 25.39 | **35.34** | **43.10** |
>
> We notice that the voxel missing in the small & rare foreground objects is much less severe than in the background (vegetation, manmade) and large objects (construction vehicle) during compression ratio increase, while these small targets are more important for safe driving. This should help address some of your safety concerns.
>
> Next we show the forecast performance as a function of VAE for different compression rates. First, the results of CFM based on latent with different compression ratios are shown below. For time considerations, all CFMs here are trained as 200 epochs.
>
> | Compression | 1&nbsp;s&nbsp;IoU | 1&nbsp;s&nbsp;mIoU| 2&nbsp;s&nbsp;IoU | 2&nbsp;s&nbsp;mIoU | 3&nbsp;s&nbsp;IoU | 3&nbsp;s&nbsp;mIoU | **Avg IoU** | **Avg mIoU** |
> |-------------|----------------:|-----------------:|----------------:|-----------------:|----------------:|-----------------:|-----------:|-------------:|
> | ×32  | 38.40 | 28.96 | 28.54 | 18.52 | 23.14 | 13.48 | 30.02 | 20.31 |
> | ×192 | **40.53** | **33.17** | **30.37** | **21.09** | **24.44** | **15.64** | **31.78** | **23.33** |
> | ×384 | 39.68 | 32.96 | 29.95 | 20.91 | 24.25 | 14.98 | 31.29 | 22.95 |
> | ×768 | 34.66| 27.10 | 24.51 | 17.30 | 22.61 | 12.95 | 27.26 | 19.11 |
>
> At the highest reconstruction rate of x32, we find that the network is not fully fitted at 200 epochs, and a longer training time may allow the network to fit better. However since this is clearly contrary to the ‘sample/training efficiency’ of this paper, we keep the training time the same for fair evaluation. For latent with higher compression multiples, the forecast performance continues to degrade. We believe that this is not only due to the fact that it retains less information on large objects, it is also because at larger compression multiples, latent becomes very sensitive to noise: we find that at multiples of x384 and x768, noise of the order of 1e-4 can crash the network.
>
> Finally, we again appreciate reviewer's comments, which makes the analysis of our article more comprehensive. We hope that our response will help to solve your problem and provides sufficient reason to raise the score.

---

### Note · Authors · 2025-08-14

Dear AC and reviewers,

Thank you for the thoughtful feedback and active discussion. During the response period, we believe we addressed all the key concerns raised by reviewers and further improved our work with new analyses and experiments, including ablations, comparisons to additional baselines, and evaluations with new metrics.

Our work introduces a LiDAR world modeling framework that achieves state-of-the-art accuracy and efficiency by pre-training a 4D motion prior from widely available autonomous driving LiDAR data. This learnt prior reduces reliance on labeled data for downstream tasks, enabling strong performance even with limited annotations. For the first time, we provide a systematic analysis of this transferability, showing that our framework surpasses widely used baselines with only 5% of labeled data.

To address concerns from the reviewers, we added new experiments and clarifications and included them in our responses. Here is a summary of them:

- On motivation and compression analysis  (Reviewer N8qa), we clarified why directly adapting RGB-based world models is impractical, and added experiments analyzing reconstruction and forecasting accuracy across compression rates, including failure cases.
- On comparisons with Occ4D and ViDAR (Reviewers qmHa and Tnf1), we clarified that Occ4D focuses only on differentiable rendering for point cloud prediction, while ViDAR cannot be fine-tuned for diverse 4D tasks and depends on UniAD with labeled data. We also added direct comparisons showing that our framework outperforms both on point cloud forecasting.
- On the effectiveness of our VAE design (Reviewer Tnf1), we added ablation studies clarifying the factors of its strong performance.
- On data transfer direction (Reviewer br6R), we justified using outdoor-to-indoor transfer by highlighting the scarcity and annotation difficulty of indoor data.
- On employing additional evaluation metrics (Reviewer qmHa), we added FID and KID for per-frame fidelity and FVD for temporal consistency, showing our method’s superiority.

We believe these additions and clarifications address the most significant concerns raised during review. While some clarifications were not explicitly acknowledged in later reviewer comments, we hope the AC will consider the full set of responses and new results in their assessment.

Thank you,

Authors

---

### Decision · Program_Chairs · 2025-09-17

**Decision:**

Accept (poster)

**Comment:**

This paper presents a LiDAR world modeling framework combining a Swin-VAE for efficient compression and conditional flow matching for temporal prediction. Pretraining on large-scale outdoor LiDAR data allows effective transfer to downstream tasks with minimal labeled data, achieving state-of-the-art results in semantic occupancy forecasting and domain adaptation. The model also demonstrates high efficiency, operating in real time with significantly fewer FLOPs than diffusion-based baselines. Reviewers appreciated the technical depth, comprehensive experiments, and originality of unifying pretraining, compression, and transfer into a single LiDAR-only world model. Strengths include clear ablations, reduced annotation needs, and systematic evaluation across tasks. The rebuttal provided valuable additions: detailed compression–forecasting trade-offs, new comparisons against Occ4D and ViDAR, and evaluations with fidelity and temporal consistency metrics (FID, KID, FVD). These addressed most concerns and strengthened the work. Remaining weaknesses include that the “foundational” claim may be overstated, the motivation for outdoor-to-indoor transfer could be clearer, and broader robustness or failure-mode analysis would further enhance the study. Nonetheless, the reviewers converged on positive scores, agreeing that the paper is technically solid, impactful, and relevant to the NeurIPS community. Overall, the submission is recommended for acceptance.